# PERTURBATION TYPE CATEGORIZATION FOR MULTIPLE $\ell_p$ BOUNDED ADVERSARIAL ROBUSTNESS

## ABSTRACT

Despite the recent advances in *adversarial training* based defenses, deep neural networks are still vulnerable to adversarial attacks outside the perturbation type they are trained to be robust against. Recent works have proposed defenses to improve the robustness of a single model against the union of multiple perturbation types. However, when evaluating the model against each individual attack, these methods still suffer significant trade-offs compared to the ones specifically trained to be robust against that perturbation type. In this work, we introduce the problem of categorizing adversarial examples based on their $\ell_p$ perturbation types. Based on our analysis, we propose PROTECTOR, a two-stage pipeline to improve the robustness against multiple perturbation types. Instead of training a single predictor, PROTECTOR first categorizes the perturbation type of the input, and then utilizes a predictor specifically trained against the predicted perturbation type to make the final prediction. We first theoretically show that adversarial examples created by different perturbation types constitute different distributions, which makes it possible to distinguish them. Further, we show that at test time the adversary faces a natural trade-off between fooling the perturbation type classifier and the succeeding predictor optimized with perturbation specific adversarial training. This makes it challenging for an adversary to plant strong attacks against the whole pipeline. In addition, we demonstrate the realization of this trade-off in deep networks by adding random noise to the model input at test time, enabling enhanced robustness against strong adaptive attacks. Extensive experiments on MNIST and CIFAR-10 show that PROTECTOR outperforms prior adversarial training based defenses by over 5%, when tested against the union of $\ell_1, \ell_2, \ell_\infty$ attacks.[1]

## 1 INTRODUCTION

There has been a long line of work studying the vulnerabilities of machine learning models to small changes in the input data. In particular, most existing works focus on $\ell_p$ bounded perturbations (Szegedy et al., 2013; Goodfellow et al., 2015). While majority of the prior work aims at achieving robustness against a single perturbation type (Madry et al., 2018; Kurakin et al., 2017; Tramèr et al., 2018; Dong et al., 2018; Zhang et al., 2019; Carmon et al., 2019), real-world deployment of machine learning models requires them to be robust against various imperceptible changes in the input, irrespective of the attack type. Prior work has shown that when models are trained to be robust against one perturbation type, such robustness typically does not transfer to attacks of a different type (Schott et al., 2018; Kang et al., 2019). As a result, recent works have proposed to develop models that are robust against the union of multiple perturbation types (Tramèr & Boneh, 2019; Maini et al., 2020). Specifically, these works consider adversaries limited by their $\ell_p$ distance from the original input for $p \in \{1, 2, \infty\}$. While these methods improve the overall robustness against multiple perturbation types, when evaluating the robustness against each individual perturbation type, the robustness of models trained by these methods is still considerably worse than those trained on a single perturbation type. Further, these methods are found sensitive to small changes in hyperparameters.

In this work, we propose an alternative view that does not require a *single* predictor to be robust against a union of perturbation types. Instead, we propose to utilize *a union of* predictors to improve the

---

[1]We will open-source the code, pre-trained models, and perturbation type datasets upon publication.

overall robustness, where each predictor is specialized to defend against certain perturbation types. In particular, we introduce the problem of categorizing adversarial examples based on their perturbation types. Based on this idea, we propose PROTECTOR, a two-stage pipeline that performs *Perturbation Type Categorization for Robustness against multiple perturbations*. Specifically, first a perturbation type classifier predicts the type of the attack. Then, among the second-level predictors, PROTECTOR selects the one that is the most robust to the predicted perturbation type to make final prediction.

We validate our approach from both theoretical and empirical aspects. First, we present theoretical analysis to show that for benign samples with the same ground truth label, their distributions become highly distinct when added with different types of perturbations, and thus can be separated. Further, we show that there exists a natural tension between attacking the top-level perturbation classifier and the second-level predictors – strong attacks against the second-level predictors make it easier for the perturbation classifier to predict the adversarial perturbation type, and fooling the perturbation classifier requires planting weaker (or less representative) attacks against the second-level predictors. As a result, even an *imperfect* perturbation classifier is sufficient to significantly improve the overall robustness of the model to multiple perturbation types.

Empirically, we show that the perturbation type classifier generalizes well on classifying adversarial examples against different adversarially trained models. Then we further compare PROTECTOR to the state-of-the-art defenses against multiple perturbations on MNIST and CIFAR-10. PROTECTOR outperforms prior approaches by over 5% against the union of the $\ell_1, \ell_2$ and $\ell_\infty$ attacks. While past work has focused on the worst case metric against all attacks, on average they suffer significant trade-offs against individual attacks. From the suite of 25 different attacks tested, the average improvement for PROTECTOR over all the attacks w.r.t. the state-of-art baseline defense is $\sim 15\%$ on both MNIST and CIFAR10. In particular, by adding random noise to the model input at test time, we further increase the tension between attacking top-level and second-level components, and bring in additional improvement of robustness against adaptive attackers. Additionally, PROTECTOR provides a modular way to integrate and update defenses against a single perturbation type.

## 2    RELATED WORK

**Adversarial examples.** The realization of the existence of adversarial examples in deep neural networks has spun active research on attack algorithms and defense proposals (Szegedy et al., 2013). Among different types of attacks (Madry et al., 2018; Hendrycks et al., 2019; Hendrycks & Dietterich, 2019; Bhattad et al., 2020), the most commonly studied ones constrain the adversarial perturbation within an $\ell_p$ region of radius $\epsilon_p$ around the original input. To improve the model robustness in the presence of such adversaries, the majority of existing defenses utilize adversarial training (Goodfellow et al., 2015), which augments the training dataset with adversarial images. Till date, different variants of the original adversarial training algorithm remain the most successful defenses against adversarial attacks (Carmon et al., 2019; Zhang et al., 2019; Wong et al., 2020; Rice et al., 2020). Other types of defenses include input transformation (Guo et al., 2018; Buckman et al., 2018) and network distillation (Papernot et al., 2016), but were rendered ineffective under stronger adversaries (He et al., 2017; Carlini & Wagner, 2017a; Athalye et al., 2018; Tramer et al., 2020). Other works have explored the relation between randomizing the inputs and adversarial examples. Tabacof & Valle (2016) analyzed the change in adversarial robustness with varying levels of noise. Hu et al. (2019) evaluated the robustness of a data point to random noise to detect adversarial examples, whereas Cohen et al. (2019) utilized randomized smoothing for certified robustness to adversarial attacks.

**Defenses against multiple perturbation types.** Recent research has been drawn towards the goal of universal adversarial robustness. Since $\ell_p$-norm bounded attacks are amongst the strongest attacks in adversarial examples literature, defending against a union of such attacks is an important step towards this end goal. Schott et al. (2018); Kang et al. (2019) showed that models that were trained for a given $\ell_p$-norm bounded attacks are not robust against attacks in a different $\ell_q$ region. Succeeding work has aimed at developing one single model that is robust against the union of multiple perturbation types. Schott et al. (2018) proposed the use of multiple variational autoencoders to achieve robustness to multiple $\ell_p$ attacks on the MNIST dataset. Tramèr & Boneh (2019) used simple aggregations of multiple adversaries to achieve non-trivial robust accuracy against the union of the $\ell_1, \ell_2, \ell_\infty$ regions. Maini et al. (2020) proposed the MSD algorithm that takes gradient steps in the union of multiple $\ell_p$ regions to improve multiple perturbation robustness. In a related line of work, Croce & Hein (2020a)

proposed a method for provable robustness against all $\ell_p$ regions for $p \geq 1$. Instead of presenting empirical results, they study the upper and lower bounds of certified robust test error on much smaller perturbation radii. Therefore, their work has a different focus, and is not directly comparable to empirical defenses studied in our work.

**Detection of adversarial examples.** Multiple prior works have focused on detecting adversarial examples (Feinman et al., 2017; Lee et al., 2018; Ma et al., 2018; Cennamo et al., 2020; Fidel et al., 2019; Yin et al., 2019a;b). However, most of these defenses have been shown to be vulnerable in the presence of adaptive adversaries (Carlini & Wagner, 2017a; Tramer et al., 2020). In comparison, our work focuses on a more challenging problem of categorizing different perturbations types. However, we show that by establishing a trade-off between fooling the perturbation classifier and the individual $\ell_p$-robust models, even an imperfect perturbation classifier is sufficient to make our pipeline robust.

## 3 PROTECTOR: PERTURBATION TYPE CATEGORIZATION FOR ROBUSTNESS

In this section, we discuss our proposed PROTECTOR approach, which performs perturbation type categorization to improve the model robustness against multiple perturbation types. We first illustrate the PROTECTOR pipeline in Figure 1, then discuss the details of each component.

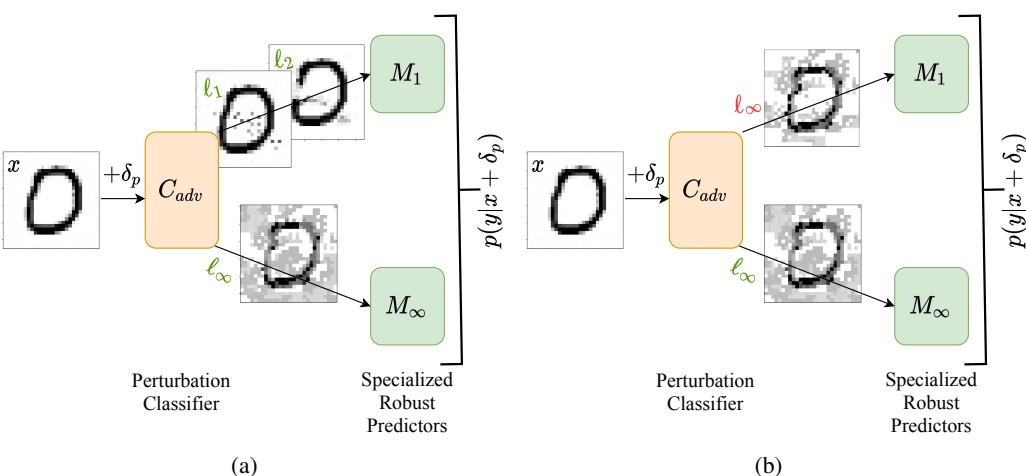

(a)                                                (b)

Figure 1: An overview of our PROTECTOR pipeline. (a) The perturbation classifier $C_{adv}$ correctly categorizes representative attacks of different types. (b) An illustration of the trade-off in Theorem 2, where an adversarial example fooling $C_{adv}$ (the $\ell_\infty$ sample marked in red) becomes weaker to attack the second-level $M_p$ models.

At a high level, PROTECTOR performs the classification task as a two-stage process. Given an input $x$, PROTECTOR first utilizes a *perturbation classifier* $C_{adv}$ to predict its adversarial perturbation type. Then, based on the $\ell_p$ attack type predicted by $C_{adv}$, PROTECTOR uses the corresponding second-level predictor $M_p$ to provide the final prediction, where $M_p$ is specially trained to be robust against the $\ell_p$ attack. Formally, let $f_\theta$ be the PROTECTOR model, then the final prediction is:

$$f_\theta(x) = M_p(x); \quad s.t. \quad p = \arg\max C_{adv}(x) \quad (1)$$

Note that when the input is a benign image, it could be classified as any perturbation type by $C_{adv}$, since all second-level predictors should achieve a high test accuracy on benign images.

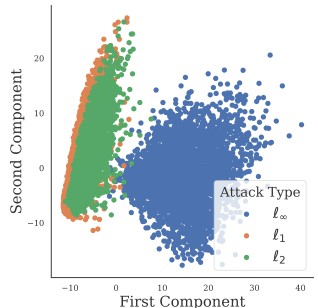

Figure 2: PCA for different types of adversarial examples on MNIST.

As shown in Figure 1, although we consider the robustness against three attack types, i.e., $\ell_1, \ell_2, \ell_\infty$ perturbations, unless otherwise specified, our perturbation classifier performs binary classification between $p = \{\{1, 2\}, \infty\}$. As will be discussed in Section 6, using two second-level predictors

achieves better overall robustness than using three second-level predictors. We hypothesize that compared to the $\ell_\infty$ adversarial examples, $\ell_1$ and $\ell_2$ attacks are harder to separate, especially when facing an adaptive adversary which aims to attack the entire pipeline. To provide an intuitive illustration, we randomly sample 10K adversarial examples generated with PGD attacks on MNIST, and visualize the results of the Principal Component Analysis (PCA) in Figure 2. We observe that the first two principal components for $\ell_1$ and $\ell_2$ adversarial examples are largely overlapping, while those for $\ell_\infty$ are clearly from a different distribution. Note that this simple visualization by no means suggest that $\ell_1$ and $\ell_2$ adversarial examples are not separable, it merely serves as a motivation.

## 4 THEORETICAL ANALYSIS

In this section, we provide a theoretical justification of our PROTECTOR framework design. First, we formally illustrate the setup of robust classification against multiple $\ell_p$ perturbation types, and we consider models trained for a binary classification task. Based on this problem setting, in Theorem 1, we show the existence of a classifier that can separate adversarial examples belonging to different perturbation types. Moreover, in Theorem 2, we show that our PROTECTOR framework naturally offers a trade-off between fooling the perturbation classifier $C_{adv}$ and the individual robust models $M_p$, thus it is extremely difficult for adversaries to stage attacks against the entire pipeline. Note that we focus on the simplified binary classification task for the convenience of theoretical analysis, but our PROTECTOR framework could improve the robustness of models trained on real-world image classification benchmarks as well, and we will discuss the empirical examination in Section 6.

### 4.1 PROBLEM SETTING

**Data distribution.** We consider a dataset of inputs sampled from the union of two multi-variate Gaussian distributions $\mathcal{D}$, such that the input-label pairs (x,y) can be described as:

$$y \overset{u.a.r}{\sim} \{-1, +1\}; \qquad x_0 \sim \mathcal{N}(y\alpha, \sigma^2), \quad x_1, \ldots, x_d \overset{i.i.d}{\sim} \mathcal{N}(y\eta, \sigma^2) \tag{2}$$

where $x = [x_0, x_1, \ldots, x_d] \in \mathcal{R}^{d+1}$ and $\eta = \frac{\alpha}{\sqrt{d}}$, such that the absolute value of the mean for any dimension is equal for inputs sampled from both the positive and the negative labels. This setting demonstrates the distinction between a feature $x_0$ that is strongly correlated with the input label, and $d$ weakly correlated features that are (independently) normally distributed with mean $y\eta$ and variance $\sigma^2$. For the purposes of this work, we assume that $\frac{\alpha}{\sigma} > 10$ ($x_0$ is strongly correlated) and $d > 100$ (remaining d features are weakly correlated, but together represent a strongly correlated feature). We adapt this problem setting from Ilyas et al. (2019), where they used a stochastic feature $x_0 = y$ with probability $p$, as opposed to a normally distributed input feature as in our case. Our results hold in their setting as well. However, our setting better represents the true data distribution, where input features are seldom stochastically flipped. More discussion could be found in Appendix A.

**Perturbation types.** We focus our discussion on adversaries constrained within a fixed $\ell_p$ region of radius $\epsilon_p$ around the original input, for $p \in \mathcal{S} = \{1, 2, \infty\}$. Such adversaries are frequently studied in existing work, primarily for finding the optimal first-order adversaries for different perturbation types. We define $\Delta_{p,\epsilon}$ as the $\ell_p$ threat model of radius $\epsilon$ and $\Delta_{\mathcal{S}} = \bigcup_{p \in \mathcal{S}} \Delta_{p,\epsilon}$. For a model $f_\theta$ parametrized over $\theta$, the objective of the adversary is to find the optimal perturbation $\delta^*$, such that:

$$\delta^* = \arg \max_{\delta \in \Delta_{\mathcal{S}}} \ell(f_\theta(x + \delta), y) \tag{3}$$

where $\ell(\cdot, \cdot)$ is the cross-entropy loss. Based on the model design in Section 3, we focus on discussing the separation of $\ell_1$ and $\ell_\infty$ in the following theorems, but our proofs could also naturally be adapted to analyze the separability of other perturbation types.

### 4.2 SEPARABILITY OF ADVERSARIAL PERTURBATIONS

Consider a standard classifier $M$ trained with the objective of correctly classifying the label of inputs $x \in \mathcal{D}$. Since the original distribution of the input data for each label is known to us, we first aim to examine how adversaries confined within different perturbation regions modify the input. The goal of the adversary is to fool the label predictor $M$, by finding the optimal perturbation $\delta_p \; \forall p \in S$. The theorem below shows that the distributions of adversarial inputs within different $\ell_p$ regions can be separated with a high accuracy, and we present the formal proof in Appendix B.

**Theorem 1** (Separability of perturbation types). *Given a binary Gaussian classifier $M$ trained on $\mathcal{D}$, consider $\mathcal{D}_p^y$ to be the distribution of optimal adversarial inputs (for a class $y$) against $M$, within $\ell_p$ regions of radius $\epsilon_p$, where $\epsilon_1 = \alpha$, $\epsilon_\infty = \alpha/\sqrt{d}$. Distributions $\mathcal{D}_p^y$ ($p \in \{1, \infty\}$) can be accurately separated by a binary Gaussian classifier $C_{adv}$ with a misclassification probability $P_e \leq 10^{-24}$.*

The proof sketch is as follows. We first calculate the optimal weights of a binary Gaussian classifier $M$ trained on $\mathcal{D}$. Accordingly, for any input $x \in \mathcal{D}$, we find the optimal adversarial perturbation $\delta_p \forall p \in \{1, \infty\}$ against $M$. We discuss how these perturbed inputs $x + \delta_p$ also follow a normal distribution, with shifted means. Finally, for data points belonging to a given classification label, we show that $C_{adv}$ is able to predict the correct perturbation type with a very low classification error. We present the formal proof in Appendix B.

### 4.3 ADVERSARIAL TRADE-OFF

In Section 4.2, we showed that the optimal perturbations corresponding to different perturbation types belong to distinct data distributions, and it is fairly easy to separate them using a simple classifier. However, in the white-box setting, the adversary has knowledge of both the perturbation classifier $C_{adv}$ and specialized robust models $M_p$ at test time. Therefore, the adversary can adapt the attack to fool the entire pipeline, instead of individual models $M_p$ alone.

Note that there are some overlapping regions among different $\ell_p$ perturbation regions. For example, every adversary could set $\delta_p = 0$ as a valid perturbation, and thus it is clearly not possible for the attack classifier $C_{adv}$ to correctly classify the attack ($\forall p \in \{1, 2, \infty\}$) in such a scenario. However, such perturbation is not useful, because all the base models can correctly classify unperturbed inputs with a high probability. In the following theorem, we examine the robustness of our PROTECTOR pipeline in the presence of such strong dynamic adversaries.

**Theorem 2** (Adversarial trade-off). *Given a data distribution $\mathcal{D}$, adversarially trained models $M_{p,\epsilon_p}$, and an attack classifier $C_{adv}$ that distinguishes perturbations of different $\ell_p$ attack types for $p \in \{1, \infty\}$, The probability of successful attack for the worst-case adversary over the entire PROTECTOR pipeline is $\boldsymbol{P_e} < 0.01$ for $\boldsymbol{\epsilon_1} = \alpha + 2\sigma$ and $\boldsymbol{\epsilon_\infty} = \frac{\alpha + 2\sigma}{\sqrt{d}}$.*

Here, the *worst-case adversary* refers to an adaptive adversary that has full knowledge of the defense strategy, and makes the strongest adversarial decision given the perturbation constraints. In Appendix C.2, we discuss how $\epsilon_1, \epsilon_\infty$ are set so that the $\ell_1$ and $\ell_\infty$ adversaries can fool $M_{\infty,\epsilon_\infty}$ and $M_{1,\epsilon_1}$ models respectively with a high success rate. Our proof sketch is as follows. We first show that when trained on $\mathcal{D}$, an adversarially robust model $M_p$ can achieve robust accuracy of greater than 99% against the attack type it was trained for. On the contrary, when subjected to attacks outside the trained perturbation region, such robust accuracy reduces to under 2%. Then, we analyze the modified distributions of the perturbed inputs by different $\ell_p$ attacks. Based on this analysis, we construct a simple decision rule for the perturbation classifier $C_{adv}$. Finally, we compute the perturbation induced by the worst-case adversary. We show that there exists a trade-off between fooling the perturbation classifier $C_{adv}$ (to allow the alternate $M_{p,\epsilon_p}$ model to make the final prediction for an $\ell_q$ attack $\forall p, q \in \{1, \infty\}; p \neq q$), and fooling the alternate $M_{p,\epsilon_p}$ model itself. Here, by "alternate" we mean that for an $\ell_q$ attack, the prediction is made by the $M_{p,\epsilon_p}$ model, where $p, q \in \{1, \infty\}; p \neq q$. We provide an illustration of the trade-off in Figure 1b, and present the formal proof in Appendix C.

## 5 TRAINING AND INFERENCE

Having motivated PROTECTOR through a toy-task in Section 4, we now scale the approach to deep neural networks for common image classification benchmarks. Specifically, following prior work on defending against multiple perturbation types, we evaluate on MNIST (LeCun et al., 2010) and CIFAR-10 (Krizhevsky, 2012) datasets. Now, we discuss the training details, a strong adaptive white-box attack against PROTECTOR, and our inference procedure against such attacks.

### 5.1 TRAINING

To train our perturbation classifier $C_{adv}$, we create a dataset that includes adversarial examples of different perturbation types. Specifically, we perform $\ell_1, \ell_2, \ell_\infty$ PGD attacks (Madry et al., 2018)

against each of the two individual $M_p$ models used in PROTECTOR. Thus the size of our dataset is 6 times that of the original MNIST and CIFAR10 datasets respectively. For the MNIST dataset, we use the $M_2, M_\infty$ models in PROTECTOR, and we use $M_1, M_\infty$ models for CIFAR10. The choice is made based on the robustness of $\{M_2, M_1\}$ models against the $\{\ell_1, \ell_2\}$ attacks respectively, as will be depicted in Table 2. As discussed in Section 3, to assign the ground truth label for training the perturbation classifier ($C_{adv}$), we find that it is sufficient to assign the same label to $\ell_1$ and $\ell_2$ attacks. In other words, $C_{adv}$ performs a binary classification between $\ell_1/\ell_2$ attacks and $\ell_\infty$ attacks.

In contrast with prior defenses against multiple perturbation types (Tramèr & Boneh, 2019; Maini et al., 2020), which require adversarial training, we find that it is sufficient to train our PROTECTOR pipeline over a static dataset (constructed as mentioned above) to achieve high robustness. Therefore, the training of our perturbation classifier is fast and stable. Specifically, using a single P100 GPU, our perturbation classifier can be trained within 5 minutes on MNIST, and within an hour on CIFAR-10. On the other hand, training state-of-the-art models robust to a single perturbation type require up to 2 days to train on the same amount of GPU power, and existing defenses against multiple perturbation types take thrice as long as the training time for a model robust to a single perturbation type.

A key advantage of PROTECTOR's design is that it can build upon existing defenses against individual perturbation types. Specifically, we leverage the adversarially trained models developed in prior work (Zhang et al., 2019; Carmon et al., 2019) as $M_p$ models in our pipeline, and the CNN architecture of $C_{adv}$ is also similar to a single $M_p$ model. More details are deferred to Appendix D.

## 5.2 ADAPTIVE ATTACKS AGAINST THE PROTECTOR PIPELINE

To generate adversarial examples against PROTECTOR, the most straightforward approach is to generate the adversarial perturbation to optimize Equation 3 using existing attack algorithms. Since the final prediction of the pipeline only depends on a single $M_p$ model, the pipeline does not allow gradient flow across the two levels, and thus makes it difficult for gradient-based adversaries to attack PROTECTOR. Therefore, besides this standard adaptive attack, in our evaluation, we also consider a stronger adaptive adversary, which utilizes a combination of the predictions from each individual second-level $M_p$ models, rather than only utilizing the predictions from a single $M_p$ model with $p = \text{argmax}\, C_{adv}(x)$ alone. Specifically, we modify $f_\theta(x)$ in Equation 3 as follows:

$$c = \text{softmax}(C_{adv}(x)); \quad f_\theta(x) = \sum_{p \in \mathcal{S}} c_p \cdot M_p(x) \tag{4}$$

where $c_p$ denotes the probability of the input $x$ being classified as the perturbation type $p$ by $C_{adv}$. We also experiment with other strategies of aggregating the predictions of different components, e.g., tuning hyper-parameters to balance among attacking $C_{adv}$ and each $M_p$ model, but these alternative methods do not perform better. Note that Equation 4 is only used for the purpose of generating adversarial examples and performing gradient-based attack optimization. For consistency throughout the paper, we still use Equation 1 to compute the model prediction at inference (final forward-propagation). We do not see any significant performance advantages of either choice at inference time, and briefly report a comparison on two attacks in Appendix H.4.

## 5.3 INFERENCE PROCEDURE AGAINST ADAPTIVE ADVERSARIES

Though training the perturbation classifier on a static dataset is sufficient to achieve robustness using existing attack approaches, we observe that the accuracy drops when PROTECTOR is presented with the stronger adaptive attacks discussed in Section 5.2. To improve the model robustness against such adversaries, we add random noise to the input before feeding it into PROTECTOR at the test time. While Hu et al. (2019) suggest that adding random noise does not help defend against adversarial inputs, it is the unique exhibition of the trade-off described in Theorem 2 that adversarial attacks against PROTECTOR, on the contrary, are highly likely to fail when added with random noise. Intuitively, the trade-off between fooling the two stages of PROTECTOR confines the adversary in a very small region for generating successful adversarial attacks.

Consider the illustrative example in Figure 3, where the input $x$ with the true label $y = 0$ is subjected to an $\ell_\infty$ attack. We assume that the $M_{\infty,\epsilon_\infty}$ model is a perfect classifier for inputs within a fixed $\epsilon_\infty$ ball. The dotted line shows the decision boundary for

the perturbation classifier $C_{adv}$, which correctly classifies inputs subjected to $\ell_\infty$ perturbations $\delta''$ as $\ell_\infty$ attacks (green), but can misclassify samples with smaller perturbations.

When the adversary adds a large perturbation $\delta''$, the prediction of $M_1$ for the resulted input $x''$ becomes wrong, but the perturbation classifier also categorizes it as an $M_\infty$ attack, thus the final prediction of PROTECTOR is still correct since it will be produced by $M_{\infty,\epsilon_\infty}$ model instead. On the other hand, when the adversary adds a small perturbation $\delta'$ to fool the perturbation classifier, adding a small amount of random noise can recover the correct prediction with a high probability. Note that every point on the boundary of the noise region (yellow circle) is correctly classified by the pipeline. In this way, adding random noise exploits an adversarial trade-off for PROTECTOR to achieve a high accuracy against adversarial examples, in the absence of adversarial training. In our implementation, we sample random noise $z \sim \mathcal{N}(0, I)$, and add $\hat{z} = \epsilon_2 \cdot z/|z|_2$ to the model input.

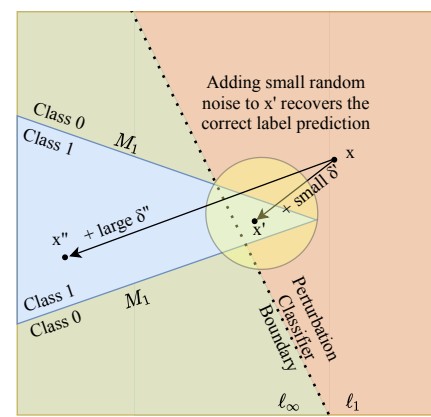

Figure 3: Illustration of the effect of random noise for generating adversarial examples. Note that the notion of small and large perturbations is only used to illustrate the scenario in Figure 3, and in general none of the perturbation regions subsumes the other.

## 6 EXPERIMENTS

In this section, we present our experiments on MNIST and CIFAR-10 datasets. We will discuss the results for both the perturbation classifier $C_{adv}$ alone, and the entire PROTECTOR pipeline.

### 6.1 EXPERIMENTAL SETUP

**Baselines.** We compare PROTECTOR with the state-of-the-art defenses against multiple perturbation types, which consider the union of $\ell_1, \ell_2, \ell_\infty$ adversaries (Tramèr & Boneh, 2019; Maini et al., 2020). For Tramèr & Boneh (2019), we compare two variants of adversarial training: (1) the **MAX** approach, where for each image, among different perturbation types, the adversarial sample that leads to the maximum increase of the model loss is augmented into the training set; (2) the **AVG** approach, where adversarial examples for all perturbation types are included for training. We also evaluate the **MSD** algorithm proposed by Maini et al. (2020), which modifies the standard PGD attack to incorporate the union of multiple perturbation types within the steepest decent itself. In addition, we also evaluate $\mathbf{M_1}, \mathbf{M_2}, \mathbf{M_\infty}$ models trained with $\ell_1, \ell_2, \ell_\infty$ perturbations separately, as described in Appendix D.

**Attack evaluation.** We evaluate our methods with the strongest attacks in the adversarial learning literature, and with adaptive attacks specifically designed for PROTECTOR (Section 5.2). First, we utilize a comprehensive suite of both gradient-based and gradient-free attacks from the Foolbox library (Rauber et al., 2017). Further, we also evaluate our method against the AutoAttack library from Croce & Hein (2020c), which achieves the state-of-art adversarial error rates against multiple recently published models. In line with prior work (Tramèr & Boneh, 2019; Maini et al., 2020), the radius of the $\{\ell_1, \ell_2, \ell_\infty\}$ perturbation regions is $\{10, 2, 0.3\}$ for the MNIST dataset and $\{12, 0.5, 0.03\}$ for the CIFAR10 dataset. We present the full details of attack algorithms in Appendix F.

Following prior work (Tramèr & Boneh, 2019; Maini et al., 2020), for both MNIST and CIFAR-10, we evaluate the models on adversarial examples generated from the first 1000 images of the test set. Our main evaluation metric is the accuracy on *all attacks*, which means that for an input image, if any of the attack algorithm in our suite could successfully fool the model, then the input is a failure case.

### 6.2 EMPIRICAL PERTURBATION OVERLAP AND CHOICE OF $\epsilon_p$

While we justify the choice of perturbation sizes in our theoretical proofs in Appendix B.4 and C.2, in this section we demonstrate the empirical agreement of the choices of perturbation sizes we make

Table 1: Studying the empirical overlap of $\ell_{p,\epsilon_p}$ attack perturbations in different $\ell_{q,\epsilon_q}$ regions for (a) MNIST $(\epsilon_1, \epsilon_2, \epsilon_\infty) = (10, 2.0, 0.3)$; (b) CIFAR-10 $(\epsilon_1, \epsilon_2, \epsilon_\infty) = (12, 0.5, 0.03)$. Each column represents the range (min - max) of $\ell_q$ norm for perturbations generated using $\ell_p$ PGD attack.

| Attack | MNIST | | | CIFAR10 | | |
|---|---|---|---|---|---|---|
| | $\ell_\infty < 0.3$ | $\ell_2 < 2.0$ | $\ell_1 < 10$ | $\ell_\infty < 0.03$ | $\ell_2 < 0.5$ | $\ell_1 < 12$ |
| PGD $\ell_\infty$ | $\leq 0.3$ | (3.67 - 6.05) | (54.8 - 140.9) | $\leq 0.03$ | (1.33 - 1.59) | (62.7 - 85.5) |
| PGD $\ell_2$ | (0.40 - 0.86) | $\leq 2.0$ | (11.2 - 24.1) | (0.037 - 0.10) | $\leq 0.05$ | (15.4 - 20.9) |
| Sparse $\ell_1$ | (0.70 - 1.0) | (2.08 - 2.92) | $\leq 10.0$ | (0.27 - 0.77) | (1.32 - 1.88) | $\leq 12.0$ |

Table 2: Worst-case accuracies against different $\ell_p$ attacks: (a) MNIST; (b) CIFAR-10. *Ours* represents PROTECTOR against the adaptive attack strategy (Section 5.2), and *Ours\** is the standard setting.

| MNIST | $M_\infty$ | $M_2$ | $M_1$ | MAX | AVG | MSD | Ours | Ours* |
|---|---|---|---|---|---|---|---|---|
| Clean Accuracy | 99.1% | 99.2% | 99.0% | 98.6% | 98.1% | 98.3% | 98.7% | 98.7% |
| $\ell_\infty$ attacks ($\epsilon = 0.3$) | 90.0% | 2.7% | 0.0% | 38.8% | 57.8% | 63.5% | 80.1% | 87.3% |
| $\ell_2$ attacks ($\epsilon = 2.0$) | 7.6% | 72.1% | 47.7% | 58.2% | 56.6% | 65.4% | 67.2% | 77.1% |
| $\ell_1$ attacks ($\epsilon = 10$) | 10.8% | 69.6% | 77.3% | 41.1% | 36.8% | 62.7% | 65.7% | 72.4% |
| All Attacks | 6.0% | 2.7% | 0.0% | 28.6% | 32.4% | 56.9% | **63.8%** | **68.5%** |

(a)

| CIFAR-10 | $M_\infty$ | $M_2$ | $M_1$ | MAX | AVG | MSD | Ours | Ours* |
|---|---|---|---|---|---|---|---|---|
| Clean accuracy | 93.3% | 91.2% | 89.3% | 81.0% | 84.6% | 81.1% | 90.8% | 90.8% |
| $\ell_\infty$ attacks ($\epsilon = 0.03$) | 59.3% | 34.8% | 35.1% | 35.0% | 39.5% | 43.7% | 57.2% | 62.1% |
| $\ell_2$ attacks ($\epsilon = 0.5$) | 64.4% | 77.2% | 71.5% | 61.8% | 65.0% | 64.5% | 66.8% | 67.1% |
| $\ell_1$ attacks ($\epsilon = 12$) | 18.8% | 34.7% | 55.0% | 34.5% | 54.1% | 51.5% | 55.9% | 56.0% |
| All attacks | 18.8% | 29.2% | 35.0% | 27.6% | 38.6% | 43.2% | **52.1%** | **54.7%** |

(b)

for our results on MNIST and CIFAR10 datasets. To measure how often adversarial perturbations of different attacks overlap, we empirically quantify the overlapping regions by attacking a benign model with PGD attacks. In Table 1 we report the range of the norm of perturbations in the alternate perturbation region for any given attack type. The observed overlap is exactly 0% in all cases and the observation is consistent across MNIST and CIFAR10 datasets. A similar analysis on attacking PROTECTOR can be found in Appendix G.

## 6.3 PERTURBATION TYPE CLASSIFICATION RESULTS OF $C_{adv}$

To examine the performance of the perturbation type classification, we evaluate $C_{adv}$ on a dataset of adversarial examples, which are generated against the six models we use as the baseline defenses in our experiments. Note that $C_{adv}$ is only trained on adversarial examples against the two $M_p$ models that are part of PROTECTOR. We observe that $C_{adv}$ transfers well across the board. First, $C_{adv}$ generalizes to adversarial examples against new models; i.e., it preserves a high accuracy, even if the adversarial examples are generated against models that are unseen for $C_{adv}$ during training. Further, $C_{adv}$ also generalizes to new attack algorithms. As discussed in Section 5.1, we only include PGD adversarial examples in our training set for $C_{adv}$. However, on adversarial examples generated by the AutoAttack library, the classification accuracy of $C_{adv}$ still holds up. In particular, the accuracy is $> 95\%$ across all the individual test sets created. These results suggest two important findings that validate our results in Theorem 1. That is, independent of **(a)** the model to be attacked; and **(b)** the algorithm for generating the optimal adversarial perturbation, the optimal adversarial images for a given $\ell_p$ region follow similar distributions. We present the full results in Appendix H.1.

## 6.4 RESULTS OF THE PROTECTOR PIPELINE

**Overall results.** In Table 2, we summarize the worst-case performance against all attacks within a given perturbation type for MNIST and CIFAR-10 datasets. In particular, 'Ours' denotes the robustness of PROTECTOR against the adaptive attacks described in Section 5.2, and 'Ours*' denotes the robustness of PROTECTOR against standard attacks based on Equation 1. The adaptive strategy

Table 3: Effect of different design choices on the CIFAR-10 dataset against PGD-based attacks. PROTECTOR(n) means the number of specialized robust predictors $M_p$ is n in the pipeline.

| | Without Noise | | With Noise | |
| | PROTECTOR(2) | PROTECTOR(3) | PROTECTOR(2) | PROTECTOR(3) |
|---|---|---|---|---|
| Clean accuracy | 90.8% | 92.1% | 90.8% | 92.2% |
| APGD $\ell_\infty$ ($\epsilon = 0.03$) | 54.7% | 50.7% | 64.8% | 56.3% |
| APGD $\ell_2$ ($\epsilon = 0.5$) | 65.0% | 64.3% | 68.8% | 69.2% |
| Sparse-PGD $\ell_1$ ($\epsilon = 12$) | 48.7% | 42.5% | 55.9% | 52.3% |

effectively reduces the overall accuracy of PROTECTOR by 2-5%, showing that incorporating the gradient and prediction information of all second-level predictors results in a stronger attack.

Despite that we evaluate PROTECTOR against a stronger adaptive adversary, in terms of the *all attacks* accuracy, PROTECTOR still outperforms all baselines by $6.9\%$ on MNIST, and $8.9\%$ on CIFAR-10. Compared to the previous state-of-the-art defense against multiple perturbation types (MSD), the accuracy gain on $\ell_\infty$ attacks is especially notable, i.e., greater than $15\%$. In particular, if we compare the performance gain on each individual attack algorithm, as shown in Appendix H.2 and H.3 for MNIST and CIFAR-10 respectively, the improvement is also significant, with an average accuracy increase of $15.5\%$ on MNIST, and $14.2\%$ on CIFAR-10. These results demonstrate that PROTECTOR considerably mitigates the trade-off in accuracy against individual attack types.

PROTECTOR retains a high accuracy on **benign images**, as opposed to past defenses that have to sacrifice the benign accuracy for the robustness on multiple perturbation types. In particular, the clean accuracy of PROTECTOR is over $6\%$ higher than such baselines on CIFAR-10, and the accuracy is similar to that of $M_p$ models trained for a single perturbation type.

**The effect of noise.** As discussed in Section 5.3, though adding random noise is not required to defend against standard attacks, it is helpful in defending against the stronger adaptive adversary against our pipeline. Specifically, in Table 3, we present the results on adversarial examples generated by PGD-based algorithms, which are amongst the strongest gradient-based attacks in the literature. We observe a consistent improvement among all attacks, increasing the accuracy by up to $10\%$.

**Different number of second-level $M_p$ predictors.** We also evaluate our PROTECTOR approach with three second-level predictors, i.e., $M_1$, $M_2$ and $M_\infty$, and we present the results in Table 3, this alternative design considerably reduces the overall accuracy of the pipeline model. We hypothesize that this happens because the $M_1$ model is already reasonably robust against the $\ell_2$ attacks, as shown in Table 2b. However, having both $M_1$ and $M_2$ models allows adaptive adversaries to find larger regions for fooling both $C_{adv}$ and $M_p$, thus hurts the overall performance against adaptive adversaries.

## 7 CONCLUSION

In this work, we propose PROTECTOR, which performs perturbation type categorization towards achieving robustness against the union of multiple perturbation types. Based on a simplified problem setup, theoretically, we demonstrate that adversarial inputs of different attack types naturally have different distributions and can be separated. We further elaborate the existence of a natural tension for any adversary trying to fool our model – between fooling the attack classifier and the specialized robust predictors. Our empirical results on MNIST and CIFAR-10 datasets complement our theoretical analysis. In particular, by posing another adversarial trade-off through the effect of random noise, our PROTECTOR pipeline outperforms existing defenses against multiple $\ell_p$ attacks by over $5\%$.

Our work serves as a stepping stone towards the goal of universal adversarial robustness, by dissecting various adversarial objectives into individually solvable pieces and combing them via PROTECTOR. Our study opens up various exciting future directions, including the new problem of perturbation categorization, extending our approach to defend attacks beyond $\ell_p$ adversarial examples, and defining sub-classes of perturbation types to further improve the overall adversarial robustness.

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

## A    PROBLEM SETTING: THEORETICAL ANALYSIS

In this section, we formally define the problem setting and motivate the distinctions made with respect to the problem studied by Ilyas et al. (2019). The classification problem consists of two tasks: **(1)** Predicting the correct class label of an adversarially perturbed (or benign) image using adversarially robust classifier $M_p$; and **(2)** Predicting the type of adversarial perturbation that the input image was subjected to, using attack classifier $C_{adv}$.

**Setup**    We consider the data to consist of inputs to be sampled from two multi-variate Gaussian distributions such that the input-label pairs (x,y) can be described as:

$$y \overset{u.a.r}{\sim} \{-1, +1\},$$
$$x_0 \sim \mathcal{N}(y\alpha, \sigma^2), \quad x_1, \ldots, x_d \overset{i.i.d}{\sim} \mathcal{N}(y\eta, \sigma^2) \tag{5}$$

where the input $x \sim \mathcal{N}(y\boldsymbol{\mu}, \boldsymbol{\Sigma}) \in \mathcal{R}^{(d+1)}$; $\eta = \alpha/\sqrt{d}$ for some positive constant $\alpha$; $\boldsymbol{\mu} = [\alpha, \eta, \ldots, \eta] \in \mathcal{R}^{+(d+1)}$ and $\boldsymbol{\Sigma} = \sigma^2 \mathbf{I} \in \mathcal{R}^{+(d+1)\times(d+1)}$. We can assume without loss of generality, that the mean for the two distributions has the same absolute value, since for any two distributions with mean $\boldsymbol{\mu}_1, \boldsymbol{\mu}_2$, we can translate the origin to $\frac{\boldsymbol{\mu}_1 + \boldsymbol{\mu}_2}{2}$. This setting demonstrates the distinction between an input feature $x_0$ that is strongly correlated with the input label and $d$ weakly correlated features that are normally distributed (independently) with mean $y\eta$ and variance $\sigma^2$ each. We adapt this setting from Ilyas et al. (2019) who used a stochastic feature $x_0 = y$ with probability $p$, as opposed to a normally distributed input feature as in our case. All our findings hold in the other setting as well, however, the chosen setting better represents true data distribution, with some features that are strongly correlated to the input label, while others that have only a weak correlation.

## B    SEPARABILITY OF PERTURBATION TYPES (THEOREM 1)

In this section, our goal is to evaluate whether the optimal perturbation confined within different $\ell_p$ balls have different distributions and whether they are separable. We do so by developing an error bound on the maximum error in classification of the perturbation types. The goal of the adversary is to fool a standard (non-robust) classifier $M$. $C_{adv}$ aims to predict the perturbation type based on **only** viewing the adversarial image, and not the delta perturbation.

First, in Appendix B.1 we define a binary Gaussian classifier that is trained on the given task. Given the weights of the binary classifier, we then identify the optimal adversarial perturbation for each of the $\ell_1, \ell_2, \ell_\infty$ attack types in Appendix B.2. In Appendix B.3 we define the difference between the adversarial input distribution for different $\ell_p$ balls. Finally, we calculate the error in classification of these adversarial input types in Appendix B.4 to conclude the proof of Theorem 1.

### B.1    BINARY GAUSSIAN CLASSIFIER

We assume for the purposes of this work that we have enough input data to be able to empirically estimate the parameters $\mu, \sigma$ of the input distribution via sustained sampling. The multivariate Gaussian representing the input data is given by:

$$p(x|y = y_i) = \frac{1}{\sqrt{(2\pi)^d |\boldsymbol{\Sigma}|}} \exp\left(-\frac{1}{2}(x - y_i.\boldsymbol{\mu})^T \boldsymbol{\Sigma}^{-1}(x - y_i.\boldsymbol{\mu})\right), \quad \forall y_i \in \{-1, 1\} \tag{6}$$

We want to find $p(y = y_i|x) \ \forall y_i \in \{-1, +1\}$. From Bayesian Decision Theory, the optimal decision rule for separating the two distributions is given by:

$$p(y = 1)p(x|y = 1) \overset{y=1}{>} p(y = -1)p(x|y = -1)$$
$$p(y = 1)p(x|y = 1) \overset{y=-1}{<} p(y = -1)p(x|y = -1) \tag{7}$$

Therefore, for two Gaussian Distributions $\mathcal{N}(\boldsymbol{\mu}_1, \boldsymbol{\Sigma}_1), \mathcal{N}(\boldsymbol{\mu}_2, \boldsymbol{\Sigma}_2)$, we have:

$$0 \overset{y=1}{\underset{<}{}} x^\top A x - 2 b^\top x + c$$
$$A = \mathbf{\Sigma}_1^{-1} - \mathbf{\Sigma}_2^{-1}$$
$$b = \mathbf{\Sigma}_1^{-1} \mu_1 - \mathbf{\Sigma}_2^{-1} \mu_2 \tag{8}$$
$$c = \mu_1^\top \mathbf{\Sigma}_1^{-1} \mu_1 - \mu_2^\top \mathbf{\Sigma}_2^{-1} \mu_2 + \log \frac{\|\Sigma_1\|}{\|\Sigma_2\|} - 2 \log \frac{p(y=1)}{p(y=-1)}$$

Substituting (6) and (7) in (8), we find that the optimal Bayesian decision rule for our problem is given by:

$$x^\top \boldsymbol{\mu} \overset{y=1}{\underset{>}{}} 0 \tag{9}$$

which means that the label for the input can be predicted with the information of the sign of $x^\top \boldsymbol{\mu}$ alone. We can define the parameters $\mathbf{W} \in \mathcal{R}^{d+1}$ of the optimal binary Gaussian classifier $M^W$, such that $\|\mathbf{W}\|_2 = 1$ as:

$$\mathbf{W}_0 = \frac{\alpha}{\sqrt{2}}, \qquad \mathbf{W}_i = \frac{\alpha}{\sqrt{2d}} \quad \forall i \in \{1, \ldots, d\}$$
$$M^W(x) = x^\top W \tag{10}$$

## B.2 Optimal Adversarial Perturbation against $M^W$

Now, we calculate the optimal perturbation $\delta$ that is added to an input by an adversary in order to fool our model. For the purpose of this analysis, we only aim to fool a model trained on the standard classification metric as discussed in Section 4 (and not an adversarially robust model). The parameters of our model are defined in (10).

The objective of any adversary $\delta \in \Delta$ is to maximize the loss of the label classifier $M^W$. We assume that the classification loss is given by $-y \times M^W(x + \delta)$. The object of the adversary is to find $\delta^*$ such that:

$$\ell(x + \delta, y; M^W) = -y \times M^W(x + \delta) = -yx^\top \mathbf{W}$$
$$\delta^* = \arg \max_{\delta \in \Delta} \ell(x + \delta, y; M^W) \tag{11}$$
$$= \arg \max_{\delta \in \Delta} -y(x + \delta)^\top \mathbf{W} = \arg \max_{\delta \in \Delta} -y\delta^\top \mathbf{W}$$

We will now calculate the optimal perturbation in the $\ell_p$ balls $\forall p \in \{1, 2, \infty\}$. For the following analyses, we restrict the perturbation region $\Delta$ to the corresponding $\ell_p$ ball of radius $\{\epsilon_1, \epsilon_2, \epsilon_\infty\}$ respectively. We also note that the optimal perturbation exists at the boundary of the respective $\ell_p$ balls. Therefore, the constraint can be re-written as :

$$\delta^* = \arg \max_{\|\delta\|_p = \epsilon_p} -y\delta^\top \mathbf{W} \tag{12}$$

We use the following properties in the individual treatment of $\ell_p$ balls:

$$\|\delta\|_p = \left( \sum_i |\delta_i|^p \right)^{\frac{1}{p}}$$
$$\partial_j \|\delta\|_p = \frac{1}{p} \left( \sum_i |\delta_i|^p \right)^{\frac{1}{p} - 1} \cdot p|\delta_j|^{p-1} \operatorname{sgn}(\delta_j) = \left( \frac{|\delta_j|}{\|\delta\|_p} \right)^{p-1} \operatorname{sgn}(\delta_j) \tag{13}$$

**p = 2**   Making use of langrange multipliers to solve (12), we have:

$$\nabla_\delta(-\delta^\top \Sigma^{-1}\mu) = \lambda\nabla_\delta(\|\delta\|_p^2 - \epsilon_p^2)$$
$$-\mathbf{W} = \lambda^{'}\|\delta\|_p\nabla_\delta(\|\delta\|_p) \tag{14}$$

Combining the results from (13) and replacing $\delta$ with $\delta_2$ we obtain :

$$-\mathbf{W} = \lambda^{'}\|\delta_2\|_2\left(\frac{|\delta_2|}{\|\delta_2\|_2}\right)\mathrm{sgn}(\delta_2)$$
$$\delta_2 = -\epsilon_2\left(\frac{\mathbf{W}}{\|\mathbf{W}\|_2}\right) = -\epsilon_2\mathbf{W} \tag{15}$$

**p = ∞**   Recall that the optimal perturbation is given by :

$$\delta^* = \arg\max_{\|\delta\|_\infty=\epsilon_\infty} -y\delta^\top\mathbf{W}$$
$$= \arg\max_{\|\delta\|_\infty=\epsilon_\infty} -y\sum_{i=0}^{d}\delta_i\mathbf{W}_i \tag{16}$$

Since $\|\delta\|_\infty = \epsilon_\infty$, we know that $\max_i|\delta_i| = \epsilon_\infty$. Therefore (16) is maximized when each $\delta_i = -y\epsilon_\infty\,\mathrm{sgn}\,\mathbf{W}_i \quad \forall i \in \{0,\ldots,d\}$. Further, since the weight matrix only contains non-negative elements ($\alpha$ is a positive constant), we can conclude that the optimal perturbation is given by:

$$\delta_\infty = -y\epsilon_\infty\mathbf{1} \tag{17}$$

**p = 1**   We attempt an analytical solution for the optimal perturbation $\delta_1$. Recall that the optimal perturbation is given by :

$$\delta^* = \arg\max_{\|\delta\|_1=\epsilon_1} -y\sum_{i=1}^{d}\delta_i\mathbf{W}_i$$
$$= \arg\max_{\|\delta\|_1=\epsilon_1} -y\delta_0\mathbf{W}_0 - y\sum_{i=1}^{d}\delta_i\mathbf{W}_i \tag{18}$$
$$= \arg\max_{\|\delta\|_1=\epsilon_1} -y\delta_0\frac{\alpha}{\sqrt{2}} - y\sum_{i=1}^{d}\delta_i\frac{\alpha}{\sqrt{2d}}$$

Since $\|\delta\|_1 = \epsilon_1$, (18) is maximized when:

$$\delta_0 = -y\epsilon_1\,\mathrm{sgn}(\alpha) = -y\epsilon_1, \qquad \delta_i = 0 \quad \forall i \in \{1\ldots d\} \tag{19}$$

**Combining the results**   From the preceding discussion, it may be noted that the new distribution of inputs within a given label changes by a different amount $\delta$ depending on the perturbation type. Moreover, if the mean and variance of the distribution of a given label are known (which implies that the corresponding true data label is also known), the optimal perturbation is independent of the input itself, and only dependent on the respective class statistics (Note that the input is still important in order to understand the true class).

### B.3   PERTURBATION CLASSIFICATION BY $C_{adv}$

In this section, we aim to verify if it is possible to accurately separate the optimal adversarial inputs crafted within different $\ell_p$ balls. For the purposes of this discussion, we only consider the problem of classifying perturbation types into $\ell_1$ and $\ell_\infty$, but the same analysis may also be extended more generally to any number of perturbation types.

We will consider the problem of classifying the correct attack label for inputs from true class $y = 1$ for this discussion. Note that the original distribution:

$$X_{true} \sim \mathcal{N}(y.\boldsymbol{\mu}, \boldsymbol{\Sigma})$$

Since the perturbation value $\delta_p$ is fixed for all inputs corresponding to a particular label, the new distribution of perturbed inputs $X_1$ and $X_\infty$ in case of $\ell_1$ and $\ell_\infty$ attacks respectively (for y = 1) is given by:

$$\begin{aligned} X_1 &\sim \mathcal{N}(\boldsymbol{\mu} + \delta_1, \boldsymbol{\Sigma}) \\ X_\infty &\sim \mathcal{N}(\boldsymbol{\mu} + \delta_\infty, \boldsymbol{\Sigma}) \end{aligned} \tag{20}$$

We now try to evaluate the conditions under which we can separate the two Gaussian distributions with an acceptable worst-case error.

### B.4 CALCULATING A BOUND ON THE ERROR

**Classification Error** A classification error occurs if a data vector x belongs to one class but falls in the decision region of the other class. That is in (7) the decision rule indicates the incorrect class. (This can be understood through the existence of outliers)

$$\begin{aligned} P_e &= \int P(\text{error}|x)p(x)dx \\ &= \int \min\left[p(y = \ell_1|x)p(x), p(y = \ell_\infty|x)p(x)\right] dx \end{aligned} \tag{21}$$

**Perturbation Size** We set the radius of the $\ell_\infty$ ball, $\epsilon_\infty = \eta$ and the radius of the $\ell_1$ ball, $\epsilon_1 = \alpha$. We further extend the discussion about suitable perturbation sizes in Appendix C.2. These values ensure that the $\ell_\infty$ adversary can make all the weakly correlated labels meaningless by changing the expected value of the adversarial input to less than 0 ($\mathbf{E}[x_i + \delta_\infty(i)] \quad \forall i > 0$), while the $\ell_1$ adversary can make the strongly correlated feature $x_0$ meaningless by changing its expected value to less than 0 ($\mathbf{E}[x_0 + \delta_1(0)]$). However, neither of the two adversaries can flip all the features together.

**Translating the axes** We can translate the axis of reference by $\left(-\mu - \left(\frac{\delta_1 + \delta_\infty}{2}\right)\right)$ and define $\boldsymbol{\mu}_{adv} = \left(\frac{\delta_1 - \delta_\infty}{2}\right)$, such that :

$$\begin{aligned} X_1 &\sim \mathcal{N}(\boldsymbol{\mu}_{adv}, \boldsymbol{\Sigma}) \\ X_\infty &\sim \mathcal{N}(-\boldsymbol{\mu}_{adv}, \boldsymbol{\Sigma}) \end{aligned} \tag{22}$$

We can once again combine this with the simplified Bayesian model in (9) to obtain the classification rule given by:

$$x^\top \boldsymbol{\mu}_{adv} \overset{p=1}{>} 0 \tag{23}$$

Combining the optimal perturbation definitions in (17) and (19) that $\boldsymbol{\mu}_{adv} = \left(\frac{\delta_1 - \delta_\infty}{2}\right) = \frac{1}{2}\left[-\epsilon_1 + \epsilon_\infty, \epsilon_\infty, \ldots, \epsilon_\infty\right]$. We can further substitute $\epsilon_1 = \alpha$ and $\epsilon_\infty = \eta = \frac{\alpha}{\sqrt{d}}$. Notice that $\boldsymbol{\mu}_{adv}(i) > 0 \ \forall i > 0$. Without loss of generality, to simplify further discussion we can flip the coordinates of $x_0$, since all dimensions are independent of each other. Therefore, $\boldsymbol{\mu}_{adv} = \frac{\alpha}{2\sqrt{d}}\left[\sqrt{d} - 1, 1, \ldots, 1\right]$. Consider a new variable $x_z$ such that:

$$x_z = x_0 \cdot \left(1 - \frac{1}{\sqrt{d}}\right) + \frac{1}{\sqrt{d}} \sum_{i=1}^{d} x_i = \frac{2}{\alpha}\left(x^\top \boldsymbol{\mu}_{adv}\right) \tag{24}$$

since each $x_i \forall i \geq 0$ is independently distributed, the new feature $x_z \sim \mathcal{N}(\mu_z, \sigma_z^2)$, where

$$\mu_z = \alpha \left(1 - \frac{1}{\sqrt{d}}\right) + \frac{1}{\sqrt{d}} \sum_{i=1}^{d} \frac{\alpha}{\sqrt{d}}$$

$$= 2\alpha - \frac{\alpha}{\sqrt{d}}$$

$$\sigma_z^2 = \sigma^2 \left(1 + \frac{1}{d} - 2\frac{1}{\sqrt{d}} + \sum_{i=1}^{d} \frac{1}{d}\right)$$

$$= \sigma^2 \left(2 + \frac{1}{d} - 2\frac{1}{\sqrt{d}}\right)$$

(25)

Therefore, the problem simplifies to calculating the probability that the meta-variable $x_z > 0$.

For $\frac{\alpha}{\sigma} > 10$ and $d > 1$, we have in the z-table, $z > 10$:

$$P_e \leq 10^{-24}$$

(26)

which suggests that the distributions are significantly distinct and can be easily separated. This concludes the proof for Theorem 1.

**Note:** We can extend the same analysis to other $\ell_p$ balls as well, but we consider the case of $\ell_1$ and $\ell_\infty$ for simplicity.

## C ROBUSTNESS OF THE PROTECTOR PIPELINE (THEOREM 2)

In the previous section, we show that it is indeed possible to distinguish between the distribution of inputs of a given class that were subjected to $\ell_1$ and $\ell_\infty$ perturbations over a standard classifier. Now, we aim to develop further understanding of the robustness of our two-stage pipeline in a dynamic attack setting with multiple labels to distinguish among. The first stage is a preliminary classifier $C_{adv}$ that classifies the perturbation type and the second stage consists of multiple models $M_p$ that were specifically trained to be robust to perturbations to the input within the corresponding $\ell_p$ norm.

First, in Appendix C.1, we calculate the optimal weights for a binary Gaussian classifier $M_p$, trained on dataset $\mathcal{D}$ to be robust to adversaries within the $\ell_p$ ball $\forall p \in \{1, \infty\}$. Based on the weights of the individual model, we fix the perturbation size $\epsilon_p$ to be only as large, as is required to fool the alternate model with high probability. Here, by 'alternate' we mean that for an $\ell_q$ attack, the prediction should be made by the $M_{p,\epsilon_p}$ model, where $p, q \in \{1, \infty\}; p \neq q$. In Appendix C.3 we calculate the robustness of individual $M_p$ models to $\ell_p$ adversaries, given the perturbation size $\epsilon_p$ as defined in Appendix C.2. In Appendix C.4, we analyze the modified distributions of the perturbed inputs after different $\ell_p$ attacks. Based on this analysis, we construct a simple decision rule for the perturbation classifier $C_{adv}$. Finally, in Appendix C.5 we determine the perturbation induced by the worst-case adversary that has complete knowledge of both $C_{adv}$ and $M_{p,\epsilon_p} \forall p \in \{1, \infty\}$. We show how there exists a trade-off between fooling the perturbation classifier (to allow the alternate $M_{p,\epsilon_p}$ model to make the final prediction), and fooling the alternate $M_{p,\epsilon_p}$ model itself.

**Perturbation Size**   We set the radius of the $\ell_\infty$ ball, $\epsilon_\infty = \eta + \zeta_\infty$ and the radius of the $\ell_1$ ball, $\epsilon_1 = \alpha + \zeta_1$, where $\zeta_p$ are some small positive constants that we calculate in Appendix C.2. These values ensure that the $\ell_\infty$ adversary can make all the weakly correlated labels meaningless by changing the expected value of the adversarial input to less than 0 ($\mathbf{E}[x_i + \delta_\infty(i)] \quad \forall i > 0$), while the $\ell_1$ adversary can make the strongly correlated feature $x_0$ meaningless by changing its expected value to less than 0 ($\mathbf{E}[x_0 + \delta_1(0)]$). However, neither of the two adversaries can flip all the features together. The exact values of $\zeta_p$ determine the exact success probability of the attacks. We defer this calculation to later when we have calculated the weights of the models $M_p$. For the following discussion, it may be assumed that $\zeta_p \to 0 \,\forall p \in \{1, \infty\}$.

## C.1 BINARY GAUSSIAN CLASSIFIER $M_p$

Extending the discussion in Appendix B.1, we now examine the learned weights of a binary Gaussian classifier $M_p$ that is trained to be robust against perturbations within the corresponding $\ell_p$ ball of radius $\epsilon_p$. The optimization equation for the classifier can be formulated as follows:

$$\min_{\mathbf{W}} \mathbb{E}\left[-yx^\top \mathbf{W}\right] + \frac{1}{2}\lambda\|\mathbf{W}\|_2^2 \tag{27}$$

where $\lambda$ is tuned in order to make the $\ell_2$ norm of the optimal weight distribution, $\|\mathbf{W}^*\|_2, = 1$. Following the symmetry argument in Lemma D.1 Tsipras et al. (2018) we extend for the binary Gaussian classifier that :

$$\mathbf{W}_i^* = \mathbf{W}_j^* = \mathbf{W_M} \quad \forall i, j \in \{1, \ldots, d\} \tag{28}$$

We deal with the cases pertaining to $p \in \{\infty, 1\}$ in this section. For both the cases, we consider existential solutions for the classifier $M_p$ to simplify the discussion. This gives us lower bounds on the performance of the optimal robust classifier. The robust objective under adversarial training can be defined as:

$$\min_{\mathbf{W}} \max_{\|\delta\|_p \leq \epsilon_p} \mathbb{E}\left[\mathbf{W}_0 \cdot (x_0 + \delta_0) + \mathbf{W_M} \cdot \sum_{i=1}^d (x_i + \delta_i)\right] + \frac{1}{2}\lambda\|\mathbf{W}\|_2^2$$

$$\min_{\mathbf{W}} \left\{-1\left(\mathbf{W_0}\alpha + d \times \mathbf{W_M}\frac{\alpha}{\sqrt{d}}\right) + \frac{1}{2}\lambda\|\mathbf{W}\|_2^2 + \max_{\|\delta\|_p \leq \epsilon_p} \mathbb{E}\left[-y\left(\mathbf{W}_0\delta_0 + \mathbf{W_M}\sum_{i=1}^d \delta_i\right)\right]\right\} \tag{29}$$

Further, since the $\lambda$ constraint only ensures that $\|\mathbf{W}^*\|_2 = 1$, we can simplify the optimization equation by substituting $\mathbf{W_0} = \sqrt{1 - d \cdot \mathbf{W_M}^2}$ as follows,

$$\min_{\mathbf{W_M}} \left\{-1\left(\alpha\sqrt{1 - d \cdot \mathbf{W_M}^2} + d \times \mathbf{W_M}\frac{\alpha}{\sqrt{d}}\right) + \max_{\|\delta\|_p \leq \epsilon_p} \mathbb{E}\left[-y\left(\delta_0\sqrt{1 - d \cdot \mathbf{W_M}^2} + \mathbf{W_M}\sum_{i=1}^d \delta_i\right)\right]\right\} \tag{30}$$

**p = $\infty$** As discussed in (17) the optimal perturbation $\delta_\infty$ is given by $-y\epsilon_\infty\mathbf{1}$. The optimization equation is simplified to:

$$\min_{\mathbf{W_M}} \left\{(\epsilon_\infty - \alpha)\sqrt{1 - d \cdot \mathbf{W_M}^2} + d \times \mathbf{W_M}\left(\epsilon_\infty - \frac{\alpha}{\sqrt{d}}\right)\right\} \tag{31}$$

Recall that $\epsilon_\infty = \frac{\alpha}{\sqrt{d}} + \zeta_\infty$. To simplify the following discussion we use the weights of a classifier trained to be robust against perturbations within the $\ell_\infty$ ball of radius $\epsilon_\infty = \frac{\alpha}{\sqrt{d}}$. The optimal solution is then given by:

$$\lim_{\zeta_\infty \to 0} \mathbf{W_M} = 0 \tag{32}$$

Therefore, the classifier weights are given by $\mathbf{W} = [\mathbf{W}_0, \mathbf{W}_1, \ldots, \mathbf{W}_d] = [1, 0, \ldots, 0]$. We also show later in Appendix C.3 that the model achieves greater than 99% accuracy against $\ell_\infty$ adversaries for the chosen values of $\zeta_\infty$.

**p = 1** We consider an analytical solution to yield optimal weights for this case. Recall from (19) that the optimal perturbation $\delta_1$ depends on the weight distribution of the classifier. Therefore, if $\mathbf{W}_0 > \mathbf{W_M}$ the optimization equation can be simplified to

$$\min_{\mathbf{W}} \left\{\mathbf{W_0}(\epsilon_1 - \alpha) - d \times \mathbf{W_M}\frac{\alpha}{\sqrt{d}} + \frac{1}{2}\lambda\|\mathbf{W}\|_2^2\right\}, \tag{33}$$

and if $\mathbf{W_M} > \mathbf{W}_0$

$$\min_{\mathbf{W}} \left\{ -\mathbf{W_0}\alpha - \mathbf{W_M}\left(\sqrt{d}\alpha - \epsilon_1\right) + \frac{1}{2}\lambda\|\mathbf{W}\|_2^2 \right\} \tag{34}$$

Recall that $\epsilon_1 = \alpha + \zeta_1$. Once again to simplify the discussion that follows we will lower bound the robust accuracy of the classifier $M_1$ by considering the optimal solution when $zeta_1 = 0$. The optimal solution is then given by:

$$\lim_{\zeta_1 \to 0} \mathbf{W_M} = 1 \tag{35}$$

For the robust classifier $M_1$, the weights $\mathbf{W} = [\mathbf{W}_0, \mathbf{W}_1, \ldots, \mathbf{W}_d] = [0, \frac{1}{\sqrt{d}}, \frac{1}{\sqrt{d}}, \ldots, \frac{1}{\sqrt{d}}]$. While this may not be the optimal solution for all values of $\zeta_1$, we are only interested in a lower bound on the final accuracy and the classifier described by weights $\mathbf{W}$ simplifies the discussion hereon. We also show later in Appendix C.3 that the model achieves greater than 99% accuracy against $\ell_1$ adversaries for the chosen values of $\zeta_1$.

## C.2    Perturbation Sizes for Fooling $M_p$ Models

Now that we exactly know the weights of the learned robust classifiers $M_1$ and $M_\infty$, we can move towards calculating values $\zeta_1$ and $\zeta_\infty$ for the exact radius of the perturbation regions for the $\ell_1$ and $\ell_\infty$ metrics. We set the radii of these regions in such a way that an $\ell_1$ adversary can fool the model $M_\infty$ with probability $\sim 98\%$ (corresponding to $z = 2$ in the z-table for normal distributions), and similarly, the success of $\ell_\infty$ attacks against the $M_1$ model is $\sim 98\%$.

Let $P_{p_1,p_2}$ represent the probability that model $M_{p_1}$ correctly classifies an adversarial input in the $\ell_{p_2}$ region. For $p_1 = \infty$ and $p_2 = 1$,

$$\begin{aligned} P_{\infty,1} &= \mathbb{P}_{x\sim\mathcal{N}(y\boldsymbol{\mu},\boldsymbol{\Sigma})}[y \cdot M_\infty(x + \delta_1) > 0] \\ &= \mathbb{P}_{x\sim\mathcal{N}(y\boldsymbol{\mu},\boldsymbol{\Sigma})}[y \cdot (x + \delta_1)^\top \mathbf{W} > 0] \\ &\geq \mathbb{P}_{x\sim\mathcal{N}(\boldsymbol{\mu},\boldsymbol{\Sigma})}[x_0 > \epsilon_1] \\ z &= \frac{\epsilon_1 - \alpha}{\sigma} = \frac{\alpha + \zeta_1 - \alpha}{\sigma} = \frac{\zeta_1}{\sigma} = 2 \\ \zeta_1 &= 2\sigma \\ \epsilon_1 &= \alpha + 2\sigma \end{aligned} \tag{36}$$

To simplify the discussion for the $M_1$ model, we define a meta-feature $x_M$ as:

$$x_M = \frac{1}{\sqrt{d}}\sum_{i=1}^{d} x_i,$$

which is distributed as :

$$x_M \sim \mathcal{N}(y\eta\sqrt{d}, \sigma^2) \sim \mathcal{N}(y\alpha, \sigma^2)$$

For $p_1 = 1$ and $p_2 = \infty$,

$$
\begin{aligned}
P_{1,\infty} &= \mathbb{P}_{x \sim \mathcal{N}(y\boldsymbol{\mu},\boldsymbol{\Sigma})}[y \cdot M_1(x + \delta_\infty) > 0] \\
&= \mathbb{P}_{x \sim \mathcal{N}(y\boldsymbol{\mu},\boldsymbol{\Sigma})}[y \cdot (x + \delta_\infty)^\top \mathbf{W} > 0] \\
&= \mathbb{P}_{x \sim \mathcal{N}(y\boldsymbol{\mu},\boldsymbol{\Sigma})}[y \cdot \frac{1}{\sqrt{d}} \sum_{i=1}^{d} (x_i + \delta_\infty(i)) > 0] \\
&= \mathbb{P}_{x \sim \mathcal{N}(y\boldsymbol{\mu},\boldsymbol{\Sigma})}[y \cdot (x_M - \sqrt{d} \cdot \epsilon_\infty) > 0] \\
&\geq \mathbb{P}_{x \sim \mathcal{N}(\boldsymbol{\mu},\boldsymbol{\Sigma})}\left[x_M > \sqrt{d} \cdot \epsilon_\infty\right]
\end{aligned}
\tag{37}
$$

$$
z = \frac{\sqrt{d} \cdot \epsilon_\infty - \alpha}{\sigma} = \frac{\alpha + \sqrt{d} \cdot \zeta_\infty - \alpha}{\sigma} = \frac{\sqrt{d} \cdot \zeta_\infty}{\sigma} = 2
$$

$$
\zeta_\infty = \frac{2\sigma}{\sqrt{d}}
$$

$$
\epsilon_\infty = \frac{\alpha + 2\sigma}{\sqrt{d}}
$$

## C.3  ROBUSTNESS OF INDIVIDUAL $M_p$ MODELS

**Additional assumptions**  We add the following assumptions: (1) the dimensionality parameter $d$ of input data is larger than 100; and (2) the ratio of the mean and variance for feature $x_0$ is greater than 10.

$$
d \geq 100, \qquad \frac{\alpha}{\sigma} \geq 10
\tag{38}
$$

We define $P_p$ as the probability that for any given input $x \sim \mathcal{N}(y\boldsymbol{\mu}, \boldsymbol{\Sigma})$, the classifier $M_p$ outputs the correct label y for the input $x + \delta_p$.

**p = $\infty$**

$$
\begin{aligned}
P_{\infty,\infty} &= \mathbb{P}_{x \sim \mathcal{N}(y\boldsymbol{\mu},\boldsymbol{\Sigma})}[y \cdot M_\infty(x + \delta_\infty) > 0] \\
&= \mathbb{P}_{x \sim \mathcal{N}(y\boldsymbol{\mu},\boldsymbol{\Sigma})}[y \cdot (x + \delta_\infty)^\top \mathbf{W} > 0] \\
&= \mathbb{P}_{x \sim \mathcal{N}(y\boldsymbol{\mu},\boldsymbol{\Sigma})}[y \cdot (x_0 + \delta_\infty(0)) > 0] \\
&\geq \mathbb{P}_{x \sim \mathcal{N}(\boldsymbol{\mu},\boldsymbol{\Sigma})}[x_0 > \epsilon_\infty]
\end{aligned}
\tag{39}
$$

$$
z = \frac{\epsilon_\infty - \alpha}{\sigma} = \frac{\alpha}{\sigma}\left(\frac{1}{\sqrt{d}} - 1\right) + \frac{2}{\sqrt{d}}
$$

using the assumptions in (38),

$$
P_{\infty,\infty} \geq 0.999
\tag{40}
$$

**p = 1**

$$
\begin{aligned}
P_{1,1} &= \mathbb{P}_{x \sim \mathcal{N}(y\boldsymbol{\mu},\boldsymbol{\Sigma})}[y \cdot M_1(x + \delta_1) > 0] \\
&= \mathbb{P}_{x \sim \mathcal{N}(y\boldsymbol{\mu},\boldsymbol{\Sigma})}[y \cdot (x + \delta_1)^\top \mathbf{W} > 0] \\
&= \mathbb{P}_{x \sim \mathcal{N}(y\boldsymbol{\mu},\boldsymbol{\Sigma})}[y \cdot \frac{1}{\sqrt{d}} \sum_{i=1}^{d} (x_i + \delta_1(i)) > 0] \\
&= \mathbb{P}_{x \sim \mathcal{N}(y\boldsymbol{\mu},\boldsymbol{\Sigma})}[y \cdot (x_M + \delta_M) > 0] \\
&\geq \mathbb{P}_{x \sim \mathcal{N}(\boldsymbol{\mu},\boldsymbol{\Sigma})}\left[x_M > \frac{\epsilon_1}{\sqrt{d}}\right]
\end{aligned}
\tag{41}
$$

$$
z = \frac{\frac{\epsilon_1}{\sqrt{d}} - \alpha}{\sigma} = \frac{\alpha}{\sigma}\left(\frac{1}{\sqrt{d}} - 1\right) + \frac{2}{\sqrt{d}}
$$

using the assumptions in (38),

$$
P_{1,1} \geq 0.999
\tag{42}
$$

## C.4 Decision rule for $C_{adv}$

We aim to provide a lower bound on the worst-case accuracy of the entire pipeline, through the existence of a simple decision tree $C_{adv}$. For given perturbation budgets $\epsilon_1$ and $\epsilon_\infty$, we aim to understand the range of values that can be taken by the adversarial input. Consider the scenarios described in Table 4 below:

Table 4: The table shows the range of the values that the mean can take depending on the decision taken by the adversary. $\mu_0^{adv}$ and $\mu_M^{adv}$ represent the new mean of the distribution of features $x_0$ and $x_M$ after the adversarial perturbation.

| Attack Type | $\mu_0^{adv}$ | | $\mu_M^{adv}$ | |
|---|---|---|---|---|
| | y = 1 | y = -1 | y = 1 | y = -1 |
| None | $\alpha$ | $-\alpha$ | $\eta\sqrt{d}$ | $-\eta\sqrt{d}$ |
| $\ell_\infty$ | $\{\alpha - \epsilon_\infty, \alpha + \epsilon_\infty\}$ | $\{-\alpha - \epsilon_\infty, -\alpha + \epsilon_\infty\}$ | $\{\eta\sqrt{d} + \epsilon_\infty d, \eta\sqrt{d} - \epsilon_\infty d\}$ | $\{-\eta\sqrt{d} + \epsilon_\infty d, -\eta\sqrt{d} - \epsilon_\infty d\}$ |
| $\ell_1$ | $\{\alpha - \epsilon_1, \alpha + \epsilon_1\}$ | $\{-\alpha - \epsilon_1, -\alpha + \epsilon_1\}$ | $\{\eta\sqrt{d} + \epsilon_1, \eta\sqrt{d} - \epsilon_1\}$ | $\{-\eta\sqrt{d} + \epsilon_1, -\eta\sqrt{d} - \epsilon_1\}$ |

Note that any adversary that moves the perturbation away from the y-axis is uninteresting for our comparison, since irrespective of a correct perturbation type prediction by $C_{adv}$, either of the two second level models naturally obtain a high accuracy on such inputs. Hence, we define the following decision rule with all the remaining cases mapped to $\ell_1$ perturbation type.

$$C_{adv}(x) = \begin{cases} 1, & \text{if} \quad ||x_0| - \alpha| < \epsilon_\infty + \frac{\alpha}{2} \\ 0, & \text{otherwise} \end{cases} \tag{43}$$

where the output 1 corresponds to the classifier predicting the presence of $\ell_\infty$ perturbation in the input, while an output of 0 suggests that the classifier predicts the input to contain perturbations of the $\ell_1$ type.

If we consider a black-box setting where the adversary has no knowledge of the classifier $C_{adv}$, and can only attack $M_p$ it is easy to see that the proposed pipeline obtains a high adversarial accuracy against the union of $\ell_1$ and $\ell_\infty$ perturbations:

Note: (1) There exists a single model that can also achieve robustness against the union of $\ell_1$ and $\ell_\infty$ perturbations, however, learning this model may be more challenging in real data settings. (2) The classifier need not be perfect.

## C.5 Trade-off between attacking $M_p$ and $C_{adv}$

To obtain true robustness it is important that the entire pipeline is robust against adversarial attacks. More specifically, in this section we demonstrate the natural tension that exists between fooling the top level attack classifier (by making an adversarial attack less representative of its natural distribution) and fooling the bottom level adversarially robust models (requiring stronger attacks leading to a return to the attack's natural distribution).

The accuracy of the pipelined model $f$ against any input-label pair $(x, y)$ sampled through some distribution $\mathcal{N}(y\boldsymbol{\mu}_{adv}, \boldsymbol{\Sigma})$ (where $\boldsymbol{\mu}_{adv}$ incorporates the change in the input distribution owing to the adversarial perturbation) is given by:

$$\begin{aligned}
\mathbb{P}\left[f(x) = y\right] &= \mathbb{P}_{x \sim \mathcal{N}(y\boldsymbol{\mu}_{adv}, \boldsymbol{\Sigma})}\left[C_{adv}(x)\right] \mathbb{P}_{x \sim \mathcal{N}(y\boldsymbol{\mu}_{adv}, \boldsymbol{\Sigma})}\left[y \cdot M_\infty(x) > 0 | C_{adv}(x)\right] \\
&\quad + (1 - \mathbb{P}_{x \sim \mathcal{N}(y\boldsymbol{\mu}_{adv}, \boldsymbol{\Sigma})}\left[C_{adv}(x)\right])\mathbb{P}_{x \sim \mathcal{N}(y\boldsymbol{\mu}_{adv}, \boldsymbol{\Sigma})}\left[y \cdot M_1(x) > 0 | \neg C_{adv}(x)\right] \\
&= \mathbb{P}_{x \sim \mathcal{N}(\boldsymbol{\mu}_{adv}, \boldsymbol{\Sigma})}\left[C_{adv}(x)\right] \mathbb{P}_{x \sim \mathcal{N}(\boldsymbol{\mu}_{adv}, \boldsymbol{\Sigma})}\left[M_\infty(x) > 0 | C_{adv}(x)\right] \\
&\quad + (1 - \mathbb{P}_{x \sim \mathcal{N}(\boldsymbol{\mu}_{adv}, \boldsymbol{\Sigma})}\left[C_{adv}(x)\right])\mathbb{P}_{x \sim \mathcal{N}(\boldsymbol{\mu}_{adv}, \boldsymbol{\Sigma})}\left[M_1(x) > 0 | \neg C_{adv}(x)\right]
\end{aligned} \tag{44}$$

$\ell_\infty$ **adversary:** To simplify the analysis, we consider loose lower bounds on the accuracy of the model $f$ against the $\ell_\infty$ adversary. Recall that the decision of the attack classifier is only dependent of the input $x_0$. Irrespective of the input features $x_i \forall i > 0$, it is always beneficial for the adversary to perturb the input by $\mu_i = -\epsilon_\infty$. However, the same does not apply for the input $x_0$. Analyzing for the

scenario when the true label $y = 1$, if the input $x_0$ lies between $\frac{\alpha}{2} - \epsilon_\infty$ of the mean $\alpha$, irrespective of the perturbation, the output of the attack classifier $C_{adv} = 1$. The $M_\infty$ model then always correctly classifies these inputs. The overall robustness of the pipeline requires analysis for the case when input lies outside $\frac{\alpha}{2} - \epsilon_\infty$ of the mean as well. However, we consider that the adversary always succeeds in such a case in order to only obtain a loose lower bound on the robust accuracy of the pipeline model $f$ against $\ell_\infty$ attacks.

$$
\begin{aligned}
\mathbb{P}\left[f(x) = y\right] &= \mathbb{P}_{x \sim \mathcal{N}(\boldsymbol{\mu}_{adv}, \boldsymbol{\Sigma})}\left[C_{adv}(x)\right] \mathbb{P}_{x \sim \mathcal{N}(\boldsymbol{\mu}_{adv}, \boldsymbol{\Sigma})}\left[M_\infty(x) > 0 | C_{adv}(x)\right] \\
&\quad + (1 - \mathbb{P}_{x \sim \mathcal{N}(\boldsymbol{\mu}_{adv}, \boldsymbol{\Sigma})}\left[C_{adv}(x)\right])\mathbb{P}_{x \sim \mathcal{N}(\boldsymbol{\mu}_{adv}, \boldsymbol{\Sigma})}\left[M_1(x) > 0 | \neg C_{adv}(x)\right] \\
&\geq \mathbb{P}_{x \sim \mathcal{N}(\boldsymbol{\mu}_{adv}, \boldsymbol{\Sigma})}\left[C_{adv}(x)\right] \mathbb{P}_{x \sim \mathcal{N}(\boldsymbol{\mu}_{adv}, \boldsymbol{\Sigma})}\left[M_\infty(x) > 0 | C_{adv}(x)\right] \\
&\geq \mathbb{P}_{x \sim \mathcal{N}(\boldsymbol{\mu}, \boldsymbol{\Sigma})}\left[|x_0 - \alpha| \leq \frac{\alpha}{2} - \epsilon_\infty\right] \\
&\geq 2\mathbb{P}_{x \sim \mathcal{N}(\boldsymbol{\mu}, \boldsymbol{\Sigma})}\left[x_0 \leq \alpha - \frac{\alpha}{2} + \epsilon_\infty\right] \\
z &= \frac{(\alpha - \frac{\alpha}{2} + \epsilon_\infty) - \alpha}{\sigma} = -\frac{\alpha}{2\sigma} + \frac{3\sigma}{2\sigma\sqrt{d}}
\end{aligned}
$$
(45)

using the assumptions in (38),

$$
\mathbb{P}\left[f(x) = y\right] \sim 0.99
$$
(46)

$\ell_1$ **adversary:** It may be noted that a trivial way for the $\ell_1$ adversary to fool the attack classifier is to return a perturbation $\delta_1 = 0$. In such a scenario, the classifier predicts that the adversarial image was subjected to an $\ell_\infty$ attack. The label prediction is hence made by the $M_\infty$ model. But we know from (40) that the $M_\infty$ model predicts benign inputs correctly with a probability $P_{\infty, \infty} > 0.99$, hence defeating the adversarial objective of misclassification. To achieve misclassification over the entire pipeline the optimal perturbation decision for the $\ell_1$ adversary when $x_0 \in \left[-\alpha - \frac{\alpha}{2} - \epsilon_1, -\alpha + \frac{\alpha}{2} + \epsilon_1\right]$ the adversary can fool the pipeline by ensuring that the $C_{adv}(x) = 1$. However, in all the other cases irrespective of the perturbation, either $C_{adv} = 0$ or the input features $x_0$ has the same sign as the label $y$. Since, $P_{1,1} > 0.99$ for the $M_1$ model, for all the remaining inputs $x_0$ the model correctly predicts the label with probability greater than 0.99 (approximate lower bound). We formulate this trade-off to elaborate upon the robustness of the proposed pipeline.

$$
\begin{aligned}
\mathbb{P}\left[f(x) = y\right] &= \mathbb{P}_{x \sim \mathcal{N}(\boldsymbol{\mu}_{adv}, \boldsymbol{\Sigma})}\left[C_{adv}(x)\right] \mathbb{P}_{x \sim \mathcal{N}(\boldsymbol{\mu}_{adv}, \boldsymbol{\Sigma})}\left[M_\infty(x) > 0 | C_{adv}(x)\right] \\
&\quad + (1 - \mathbb{P}_{x \sim \mathcal{N}(\boldsymbol{\mu}_{adv}, \boldsymbol{\Sigma})}\left[C_{adv}(x)\right])\mathbb{P}_{x \sim \mathcal{N}(\boldsymbol{\mu}_{adv}, \boldsymbol{\Sigma})}\left[M_1(x) > 0 | \neg C_{adv}(x)\right] \\
&\geq \mathbb{P}_{x \sim \mathcal{N}(\boldsymbol{\mu}, \boldsymbol{\Sigma})}\left[-\alpha - \frac{\alpha}{2} - \epsilon_1 \leq x_0 \leq -\alpha + \frac{\alpha}{2} + \epsilon_1\right] \\
&\quad + 0.999(\mathbb{P}_{x \sim \mathcal{N}(\boldsymbol{\mu}, \boldsymbol{\Sigma})}\left[x_0 < -\alpha - \frac{\alpha}{2} - \epsilon_1 \text{ or } x_0 > -\alpha + \frac{\alpha}{2} + \epsilon_1\right]) \\
&\geq 0.999(\mathbb{P}_{x \sim \mathcal{N}(\boldsymbol{\mu}, \boldsymbol{\Sigma})}\left[x_0 < -\alpha - \frac{\alpha}{2} - \epsilon_1 \text{ or } x_0 > -\alpha + \frac{\alpha}{2} + \epsilon_1\right])
\end{aligned}
$$
(47)

using the assumptions in (38),

$$
\mathbb{P}\left[f(x) = y\right] \sim 0.99
$$
(48)

This concludes the proof for Theorem 2, showing that an adversary can hardly stage successful attacks on the entire pipeline and faces a natural tension between attacking the label predictor and the attack classifier. Finally, we emphasize that the shown accuracies are lower bounds on the actual robust accuracy, and the objective of this analysis is not to find the optimal solution to the problem of multiple perturbation adversarial training, but to expose the existence of the trade-off between attacking the two stages of the pipeline.

## D   MODEL ARCHITECTURE

**Second-level $M_p$ models.**   A key advantage of our PROTECTOR design is that we can build upon existing defenses against individual perturbation type. Specifically, for MNIST, we use the same CNN architecture as Zhang et al. (2019) for our $M_p$ models, and we train these models using their proposed TRADES loss. For CIFAR-10, we use the same training setup and model architecture as Carmon et al. (2019), which is based on a robust self-training algorithm that utilizes unlabeled data to improve the model robustness.

**Perturbation classifier $C_{adv}$.**   For both MNIST and CIFAR-10 datasets, the architecture of the perturbation classifier $C_{adv}$ is similar to the individual $M_p$ models. Specifically, for MNIST, we use the CNN architecture in Zhang et al. (2019) with four convolutional layers, followed by two fully-connected layers. For CIFAR-10, $C_{adv}$ is a WideResNet (Zagoruyko & Komodakis, 2016) model with depth 28 and widening factor of 10 (WRN-28-10).

## E   TRAINING DETAILS

### E.1   SPECIALIZED ROBUST PREDICTORS $M_p$

**MNIST.**   We use the Adam optimizer (Kingma & Ba, 2015) to train our models along with a piece-wise linearly varying learning rate schedule (Smith, 2018) to train our models with maximum learning rate of $10^{-3}$. The base models $M_1, M_2, M_\infty$ are trained using the TRADES algorithm for 20 iterations, and step sizes $\alpha_1 = 2.0$, $\alpha_2 = 0.3$, and $\alpha_\infty = 0.05$ for the $\ell_1, \ell_2, \ell_\infty$ attack types within perturbation radii $\epsilon_1 = 10.0$, $\epsilon_2 = 2.0$, and $\epsilon_\infty = 0.3$ respectively.[2]

**CIFAR10.**   The individual $M_p$ models are trained to be robust against $\{\ell_\infty, \ell_1, \ell_2\}$ perturbations of $\{\epsilon_\infty, \epsilon_1, \epsilon_2\} = \{0.003, 12.0, 0.05\}$ respectively. For CIFAR10, the attack step sizes $\{\alpha_\infty, \alpha_1, \alpha_2\} = \{0.005, 2.0, 0.1\}$ respectively. The training of the individual $M_p$ models is directly based on the work of Carmon et al. (2019).

### E.2   PERTURBATION CLASSIFIER $C_{adv}$

**MNIST.**   We use a learning rate of 0.01 and Adam optimizer for 10 epochs, with linear rate decay to 0.001 between the fourth epoch and the tenth epoch. The batch size is set to 100 for all experiments.

**CIFAR10.**   We use a learning rate of 0.01 and SGD optimizer for 5 epochs, with linear rate decay to 0.001 between the fourth epoch and the tenth epoch. The batch size is set to 100 for all experiments.

**Creating the Adversarial Perturbation Dataset.**   We create a static dataset of adversarially perturbed images and their corresponding attack label for training the perturbation classifier $C_{adv}$. For generating adversarial images, we perform weak adversarial attacks that are faster to compute. In particular, we perform 10 iterations of the PGD attack. For MNIST, the attack step sizes $\{\alpha_\infty, \alpha_1, \alpha_2\} = \{0.05, 2.0, 0.3\}$ respectively. For CIFAR10, the attack step sizes $\{\alpha_\infty, \alpha_1, \alpha_2\} = \{0.005, 2.0, 0.1\}$ respectively. Note that we perform the Sparse-$\ell_1$ or the top-k PGD attack for the $\ell_1$ perturbation ball, as introduced by Tramèr & Boneh (2019). We set the value of k to 10, that is we move by a step size $\frac{\alpha_1}{k}$ in each of the top 10 directions with respect to the magnitude of the gradient.

## F   ATTACKS USED FOR EVALUATION

A description of all the attacks used for evaluation of the models is presented here. From the Foolbox library(Rauber et al., 2017), apart from $\ell_1, \ell_2$ and $\ell_\infty$ PGD adversaries, we also evaluate the following attacks for different perturbation types.

(1) For $\ell_1$ perturbations, we include the Salt & Pepper Attack (SAPA) (Rauber et al., 2017) and Pointwise Attack (PA) (Schott et al., 2018).

---

[2]We use the Sparse $\ell_1$ descent Tramèr & Boneh (2019) for the PGD attack in the $\ell_1$ constraint.

Table 5: **Vanilla Model:** Empirical overlap of $\ell_{p,\epsilon_p}$ attack perturbations in different $\ell_{q,\epsilon_q}$ regions for (a) MNIST $(\epsilon_1, \epsilon_2, \epsilon_\infty) = (10, 2.0, 0.3)$; (b) CIFAR-10 $(\epsilon_1, \epsilon_2, \epsilon_\infty) = (12, 0.5, 0.03)$. Each column represents the range (min - max) of $\ell_q$ norm for perturbations generated using $\ell_p$ PGD attack.

| Attack | MNIST | | | CIFAR10 | | |
|---|---|---|---|---|---|---|
| | $\ell_\infty < 0.3$ | $\ell_2 < 2.0$ | $\ell_1 < 10$ | $\ell_\infty < 0.03$ | $\ell_2 < 0.5$ | $\ell_1 < 12$ |
| PGD $\ell_\infty$ | $\leq 0.3$ | (3.67 - 6.05) | (54.8 - 140.9) | $\leq 0.03$ | (1.33 - 1.59) | (62.7 - 85.5) |
| PGD $\ell_2$ | (0.40 - 0.86) | $\leq 2.0$ | (11.2 - 24.1) | (0.037 - 0.10) | $\leq 0.05$ | (15.4 - 20.9) |
| Sparse $\ell_1$ | (0.70 - 1.0) | (2.08 - 2.92) | $\leq 10.0$ | (0.27 - 0.77) | (1.32 - 1.88) | $\leq 12.0$ |

Table 6: **PROTECTOR:** Empirical overlap of $\ell_{p,\epsilon_p}$ attack perturbations in different $\ell_{q,\epsilon_q}$ regions for (a) MNIST $(\epsilon_1, \epsilon_2, \epsilon_\infty) = (10, 2.0, 0.3)$; (b) CIFAR-10 $(\epsilon_1, \epsilon_2, \epsilon_\infty) = (12, 0.5, 0.03)$. Each column represents the range (min - max) of $\ell_q$ norm for perturbations generated using $\ell_p$ PGD attack.

| Attack | MNIST | | | CIFAR10 | | |
|---|---|---|---|---|---|---|
| | $\ell_\infty < 0.3$ | $\ell_2 < 2.0$ | $\ell_1 < 10$ | $\ell_\infty < 0.03$ | $\ell_2 < 0.5$ | $\ell_1 < 12$ |
| PGD $\ell_\infty$ | $\leq 0.3$ | (5.03-6.12) | (100.40-138.52) | $\leq 0.03$ | (1.46-1.69) | (73.15-93.26) |
| PGD $\ell_2$ | (0.35-0.95) | $\leq 2.0$ | (17.06-27.88) | (0.036-0.29) | $\leq 0.05$ | (5.83-21.21) |
| Sparse $\ell_1$ | (0.81-1.0) | (2.13-2.98) | $\leq 10.0$ | (0.42-1.0) | (1.50-2.91) | $\leq 12.0$ |

(2) For $\ell_2$ perturbations, we include the Gaussian noise attack (Rauber et al., 2017), Boundary Attack (Brendel et al., 2018), DeepFool (Moosavi-Dezfooli et al., 2016), DDN attack (Rony et al., 2019), and C&W attack (Carlini & Wagner, 2017b).

(3) For $\ell_\infty$ perturbations, we include FGSM attack (Goodfellow et al., 2015) and the Momentum Iterative Method (Dong et al., 2018).

From the AutoAttack library from Croce & Hein (2020c), we make use of all the three variants of the Adaptive PGD attack (APGD-CE, APGD-DLR, APGD-T) (Croce & Hein, 2020c) along with the targeted and standard version of Fast Adaptive Boundary Attack (FAB, FAB-T) (Croce & Hein, 2020b) and the Square Attack (Andriushchenko et al., 2020). We utilize the AA$^+$ version for strong attacks.

**Attack Hyperparameters**   For the attacks in the Foolbox and AutoAtack libraries we use the default parameter setting in the strongest available mode (such as AA$^+$). For the custom PGD attacks, we evaluate the models with 10 restarts and 200 iterations of the PGD attack. The step size of the $\{\ell_\infty, \ell_1, \ell_2\}$ PGD attacks are set as follows: For MNIST, the attack step sizes $\{\alpha_\infty, \alpha_1, \alpha_2\} = \{0.01, 1.0, 0.1\}$ respectively. For CIFAR10, the attack step sizes $\{\alpha_\infty, \alpha_1, \alpha_2\} = \{0.003, 1.0, 0.02\}$ respectively.

Further, similar to Tramèr & Boneh (2019); Maini et al. (2020) we evaluate our models on the first 1000 images of the test set of MNIST and CIFAR-10, since many of the attacks employed are extremely computationally expensive and slow to run. Specifically, on a single GPU, the entire evaluation for a single model against all the attacks discussed with multiple restarts will take nearly 1 month, and is not feasible.

# G   EMPIRICAL PERTURBATION OVERLAP

Following Section 6.2, we also present results on the perturbation overlap when we attack PROTECTOR with PGD attacks. To contrast the results with that of attacking a vanilla model, we also present the table in the main paper for convenience. It is noteworthy that the presence of a perturbation classifier forces the adversaries to generate such attacks that increase the norm of the perturbations in alternate $\ell_q$ region. Secondly, we also observe that in the case of CIFAR10, the $\ell_2$ PGD attack has a large overlap with the $\ell_1$ norm of radius 10. However, recall that in case of $\ell_2$ attacks for CIFAR10, both the base models $M_1$ and $M_\infty$ were satisfactorily robust. Hence, the attacker has no incentive to reduce the perturbation radius for an $\ell_q$ norm since the perturbation classifier only performs a binary classification between $\ell_1$ and $\ell_\infty$ attacks. The results can be observed in Tables 5 and 6.

Table 7: Perturbation type classification accuracy for different perturbation types. Note that the perturbation classifier $C_{adv}$ is only trained on adversarial examples against two $M_p$ models. Each column represent the model used to create transfer-based attack via the attack type in the corresponding row. The represented accuracy is an aggregate over 1000 randomly sampled attacks of the $\ell_\infty, \ell_2, \ell_1$ types for the corresponding algorithms (and datasets).

|  | $M_\infty$ | $M_2$ | $M_1$ | MAX | AVG | MSD |
|---|---|---|---|---|---|---|
| MNIST-PGD | 100.0% | 100.0% | 99.3% | 99.0% | 99.6% | 99.1% |
| MNIST-AutoAttack | 100.0% | 100.0% | 99.0% | 99.5% | 100.0% | 100.0% |
| CIFAR10-PGD | 99.90% | 99.50% | 100.0% | 100.0% | 98.70% | 95.70% |
| CIFAR10-AutoAttack | 99.90% | 99.90% | 100.0% | 100.0% | 99.70% | 99.70% |

## H    BREAKDOWN OF COMPLETE EVALUATION

In this section, we present the results of the perturbation type classifier $C_{adv}$ against transfer adversaries. We also present the breakdown results of the adversarial robustness of baseline approaches and our PROTECTOR pipeline against all the attacks that we tried, and also report the worst case performance against the union of all attacks.

### H.1    ROBUSTNESS OF $C_{adv}$

The results for the robustness of the perturbation classifier $C_{adv}$ in the presence of adaptive adversaries is presented in Table 7. Note that $C_{adv}$ transfers well across the board, even if the adversarial examples are generated against new models that are unseen for $C_{adv}$ during training, achieving extremely high test accuracy. Further, even if the adversarial attack was generated by a different algorithm such as from the AutoAttack library, the transfer success of $C_{adv}$ still holds up. In particular, the obtained accuracy is $> 95\%$ across all the individual test sets created. The attack classification accuracy is in general highest against those generated by attacking $M_1$ or $M_\infty$ for CIFAR10, and $M_2$ or $M_\infty$ for MNIST. This is an expected consequence of the nature of generation of the static dataset for training the perturbation classifier $C_{adv}$ as described in Section 5.1.

### H.2    MNIST

In Table 8, we provide a breakdown of the adversarial accuracy of all the baselines, individual $M_p$ models and the PROTECTOR method, with both the adaptive and standard attack variants on the MNIST dataset. PROTECTOR outperforms prior baselines by $6.9\%$ on the MNIST dataset. It is important to note that PROTECTOR shows significant improvements against most attacks in the suite. Compared to the previous state-of-the-art defense against multiple perturbation types (MSD), if we compare the performance gain on each individual attack algorithm, the improvement is also significant, with an average accuracy increase of $15.5\%$ on MNIST dataset. These results demonstrate that PROTECTOR considerably mitigates the trade-off in accuracy against individual attack types.

### H.3    CIFAR-10

In Table 9, we provide a breakdown of the adversarial accuracy of all the baselines, individual $M_p$ models and the PROTECTOR method, with both the adaptive and standard attack variants on the CIFAR10 dataset. PROTECTOR outperforms prior baselines by $8.9\%$. Once again, note that PROTECTOR shows significant improvements against most attacks in the suite. Compared to the previous state-of-the-art defense against multiple perturbation types (MSD), if we compare the performance gain on each individual attack algorithm, the improvement is significant, with an average accuracy increase of $14.2\%$ on. These results demonstrate that PROTECTOR considerably mitigates the trade-off in accuracy against individual attack types.

Further, PROTECTOR also retains a higher accuracy on benign images, as opposed to past defenses that have to sacrifice the benign accuracy for the robustness on multiple perturbation types. In particular, the clean accuracy of PROTECTOR is over $6\%$ higher than such existing defenses on CIFAR-10, and the accuracy is close to $M_p$ models trained for a single perturbation type.

| | $M_\infty$ | $M_2$ | $M_1$ | MAX | AVG | MSD | Ours | Ours* |
|---|---|---|---|---|---|---|---|---|
| Benign Accuracy | 99.2% | 98.7% | 98.8% | 98.6% | 99.1% | 98.3% | 98.7% | 98.7% |
| $\ell_\infty$ attacks ($\epsilon = 0.3$) | | | | | | | | |
| FGSM | 96.3% | 87.7% | 12.2% | 81.2% | 85.4% | 83.0% | 94.8% | 94.8% |
| PGD-Foolbox | 95.5% | 27.5% | 0.4% | 67.8% | 76.0% | 76.2% | 86.7% | 95.1% |
| MIM | 94.1% | 52.5% | 1.1% | 70.6% | 76.7% | 74.2% | 86.8% | 96.7% |
| APGD-CE | 91.6% | 3.8% | 0.0% | 41.0% | 59.0% | 65.3% | 86.9% | 91.6% |
| APGD-DLR | 91.9% | 7.7% | 0.0% | 43.8% | 61.9% | 65.8% | 91.8% | 91.5% |
| APGD-T | 91.8% | 2.8% | 0.0% | 39.6% | 58.9% | 64.5% | 90.5% | 92.5% |
| FAB | 91.8% | 5.5% | 0.0% | 51.0% | 65.7% | 66.8% | 95.6% | 97.7% |
| FAB-T | 92.6% | 4.9% | 0.0% | 48.7% | 64.2% | 65.5% | 95.2% | 98.0% |
| SQUARE-T | 90.1% | 7.3% | 0.0% | 46.0% | 64.9% | 68.0% | 97.2% | 97.3% |
| $\ell_2$ attacks ($\epsilon = 2.0$) | | | | | | | | |
| PGD-$\ell_2$ | 78.2% | 73.6% | 50.6% | 62.6% | 65.9% | 71.0% | 73.5% | 74.3% |
| PGD-Foolbox | 95.4% | 83.2% | 60.2% | 74.3% | 79.2% | 79.0% | 83.7% | 83.3% |
| Gaussian Noise | 99.4% | 98.5% | 98.6% | 98.6% | 99.1% | 98.1% | 98.5% | 98.5% |
| DeepFool | 94.4% | 88.9% | 71.0% | 81.0% | 86.7% | 81.7% | 87.7% | 88.0% |
| DDN | 41.7% | 75.9% | 53.2% | 62.1% | 64.7% | 70.3% | 92.4% | 94.7% |
| CWL2 | 51.7% | 76.3% | 54.7% | 63.6% | 65.2% | 71.5% | 86.8% | 98.2% |
| APGD-CE | 78.5% | 74.2% | 50.8% | 61.9% | 64.9% | 69.7% | 75.7% | 76.4% |
| APGD-DLR | 79.6% | 75.4% | 53.9% | 63.2% | 65.1% | 70.8% | 76.5% | 78.2% |
| APGD-T | 80.4% | 73.5% | 48.0% | 61.2% | 63.8% | 69.4% | 74.5% | 76.8% |
| FAB | 11.6% | 77.5% | 54.9% | 63.4% | 64.3% | 69.5% | 98.0% | 98.0% |
| FAB-T | 13.2% | 74.9% | 49.4% | 62.3% | 63.6% | 69.0% | 86.3% | 96.3% |
| SQUARE-T | 25.9% | 81.9% | 67.1% | 71.9% | 71.4% | 73.7% | 96.5% | 96.6% |
| $\ell_1$ attacks ($\epsilon = 10.0$) | | | | | | | | |
| PGD-$\ell_1$ | 65.6% | 73.6% | 78.2% | 52.0% | 57.1% | 65.5% | 69.3% | 79.4% |
| Salt & Pepper | 67.1% | 95.4% | 96.2% | 93.9% | 88.2% | 91.2% | 95.2% | 95.4% |
| Pointwise Attack | 15.6% | 84.1% | 88.4% | 66.0% | 46.2% | 78.3% | 79.5% | 89.5% |
| FAB-T | 19.1% | 71.5% | 77.9% | 43.9% | 41.1% | 67.6% | 82.1% | 95.0% |

Table 8: Attack-wise breakdown of adversarial robustness on the MNIST dataset. *Ours* represents the PROTECTOR method against the adaptive attack strategy described in Section 5.2, and *Ours\** represents the standard attack setting.

| | $M_\infty$ | $M_2$ | $M_1$ | MAX | AVG | MSD | Ours | Ours* |
|---|---|---|---|---|---|---|---|---|
| Benign Accuracy | 89.4% | 93.9% | 89.3% | 81.0% | 84.6% | 81.1% | 90.8% | 90.8% |
| $\ell_\infty$ attacks ($\epsilon = 0.03$) | | | | | | | | |
| FGSM | 70.2% | 62.7% | 51.3% | 53.3% | 50.4% | 53.0% | 80.4% | 72.1% |
| PGD-Foolbox | 69.3% | 53.9% | 47.0% | 52.3% | 49.0% | 52.0% | 68.8% | 68.9% |
| MIM | 66.1% | 49.8% | 43.0% | 49.0% | 44.5% | 49.6% | 67.4% | 72.1% |
| APGD-CE | 61.9% | 35.6% | 35.7% | 38.7% | 41.0% | 46.2% | 64.8% | 63.8% |
| APGD-DLR | 61.1% | 38.0% | 37.7% | 39.6% | 43.0% | 46.4% | 62.5% | 63.2% |
| APGD-T | 59.4% | 34.8% | 35.1% | 36.8% | 39.7% | 43.8% | 62.0% | 62.1% |
| FAB | 60.5% | 38.2% | 36.5% | 40.9% | 40.4% | 44.4% | 87.1% | 88.1% |
| FAB-T | 60.0% | 35.8% | 35.5% | 40.8% | 40.2% | 44.0% | 78.2% | 83.1% |
| SQUARE-T | 67.2% | 57.4% | 50.5% | 51.7% | 50.9% | 51.5% | 86.1% | 84.1% |
| $\ell_2$ attacks ($\epsilon = 0.5$) | | | | | | | | |
| PGD-$\ell_2$ | 66.4% | 77.5% | 72.4% | 64.5% | 67.8% | 66.2% | 67.0% | 68.3% |
| PGD-Foolbox | 68.7% | 78.1% | 72.7% | 64.9% | 67.9% | 66.3% | 69.8% | 70.4% |
| Gaussian Noise | 89.4% | 93.9% | 89.1% | 81.7% | 84.7% | 82.3% | 89.3% | 89.3% |
| DeepFool | 72.9% | 79.4% | 73.0% | 64.3% | 67.3% | 65.6% | 72.8% | 73.1% |
| DDN | 66.8% | 77.5% | 72.6% | 64.6% | 67.7% | 66.2% | 83.6% | 85.2% |
| CWL2 | 67.2% | 77.5% | 72.0% | 63.2% | 71.5% | 65.1% | 88.4% | 68.3% |
| APGD-CE | 66.2% | 77.4% | 72.3% | 64.3% | 67.2% | 66.1% | 68.8% | 70.2% |
| APGD-DLR | 65.6% | 77.6% | 71.9% | 62.9% | 66.0% | 65.3% | 68.8% | 69.3% |
| APGD-T | 64.8% | 77.3% | 71.5% | 62.1% | 65.5% | 64.5% | 67.8% | 68.6% |
| FAB | 65.7% | 77.8% | 71.8% | 62.5% | 65.6% | 64.6% | 89.9% | 88.2% |
| FAB-T | 65.0% | 77.4% | 71.7% | 62.7% | 65.6% | 64.5% | 85.2% | 85.6% |
| SQUARE-T | 80.8% | 86.2% | 82.0% | 72.0% | 77.3% | 72.1% | 89.4% | 88.6% |
| $\ell_1$ attacks ($\epsilon = 12.0$) | | | | | | | | |
| PGD-$\ell_1$ | 20.6% | 38.2% | 55.9% | 41.2% | 54.7% | 53.4% | 55.9% | 56.3% |
| Salt & Pepper | 66.9% | 81.1% | 80.2% | 73.8% | 80.2% | 74.2% | 74.4% | 72.0% |
| Pointwise Attack | 50.4% | 70.6% | 74.6% | 54.9% | 75.1% | 69.6% | 78.9% | 70.1% |
| FAB-T | 28.3% | 36.8% | 56.4% | 41.3% | 60.2% | 52.2% | 70.5% | 70.1% |

Table 9: Attack-wise breakdown of adversarial robustness on the CIFAR-10 dataset. *Ours* represents the PROTECTOR method against the adaptive attack strategy described in Section 5.2, and *Ours\** represents the standard attack setting.

Table 10: Comparison between using a 'softmax' based aggregation of predictions from different specialized models versus using the prediction from the model corresponding to the most likely attack (only at inference time). Results are presented for APGD $\ell_2, \ell_\infty$ attacks on the CIFAR10 dataset.

| Attack | Max-approach (Eq. 1) | Softmax-approach (Eq. 4) |
|---|---|---|
| APGD-CE $\ell_2$ ($\epsilon_2 = 0.5$) | 75.7% | 75.6% |
| APGD-DLR $\ell_2$ ($\epsilon_2 = 0.5$) | 76.5% | 76.7% |
| APGD-CE $\ell_\infty$ ($\epsilon_\infty = 0.03$) | 86.9% | 86.9% |
| APGD-DLR $\ell_\infty$ ($\epsilon_\infty = 0.03$) | 91.8% | 91.2% |

### H.4 Aggregating predictions from different $M_p$ at Inference

In all our experiments in this work the adversary constructs adversarial examples using the softmax based adaptive strategy for aggregating predictions from different $M_p$ models, as described in Equation 4 for the column 'Ours' and using the 'max' strategy (Equation 1) for results described in the column 'Ours*'

However, for consistency of our defense strategy irrespective of the attacker's strategy, the defender only utilizes predictions from the specialized model $M_p$ corresponding to the most-likely attack (Equation 1) to provide the final prediction (only forward propagation) for generated adversarial examples. In our evaluation, we found a negligible impact of changing this aggregation to the 'softmax' strategy for aggregating the predictions. For example, we show representative results in case of the APGD ($\ell_\infty, \ell_2$) attacks on the CIFAR10 dataset in Table 10.

