# OpenReview forum: "Perturbation Type Categorization for Multiple $\ell_p$ Bounded Adversarial Robustness"
_ICLR.cc/2021/Conference — Reject_

### Official Review · AnonReviewer4 · 2020-10-27
**assumptions on distribution unrealistic**

**Rating:** 4
**Confidence:** 3

**Review:**

In theoretical analysis, the authors conclude with an Gaussian assumption on the data distribution.

This assumption is very restrictive, and the corresponding conclusion does not provide a general view of what is happening for real datasets. Gaussian condition on dataset is very unrealistic, and its theoretical analysis should only be considered as dealing with a toy model.

Authors should remove this Gaussian assumption, or at least give strong reasons in plain natural language why Gaussianity can represent natural datasets in the adversarial study.

---

> ### Author Response · Authors · 2020-11-12
> **Justification‌ ‌of‌ ‌Assumptions‌ ‌made**
>
> Thank‌ ‌you‌ ‌for‌ ‌your‌ ‌comment.‌ ‌We‌ ‌would‌ ‌like‌ ‌to‌ ‌point‌ ‌out‌ ‌that‌ ‌in‌ ‌the‌ ‌adversarial‌ ‌machine‌ ‌learning‌ ‌
> literature,‌ ‌people‌ ‌have‌ ‌relied‌ ‌on‌ ‌certain‌ ‌assumptions‌ ‌to‌ ‌build‌ ‌theoretical‌ ‌frameworks.‌ ‌Though‌ ‌
> these‌ ‌theoretical‌ ‌frameworks‌ ‌are‌ ‌based‌ ‌on‌ ‌simplified‌ ‌assumptions,‌ ‌they‌ ‌motivate‌ ‌empirical‌ ‌
> approaches‌ ‌that‌ ‌work‌ ‌in‌ ‌practice.‌ ‌Specifically,‌ ‌the‌ ‌Gaussian‌ ‌assumption‌ ‌has‌ ‌been‌ ‌made‌ ‌in‌ ‌a‌ ‌
> number‌ ‌of‌ ‌prior‌ ‌work‌ ‌on‌ ‌adversarial‌ ‌machine‌ ‌learning.‌ ‌In‌ ‌the‌ ‌following,‌ ‌we‌ ‌list‌ ‌a‌ ‌few‌ ‌papers‌ ‌
> published‌ ‌in‌ ‌recent‌ ‌machine‌ ‌learning‌ ‌conferences:‌   ‌
>
> [[NeurIPS’19]‌ ‌Ilyas‌ ‌et‌ ‌al.,‌ ‌Adversarial‌ ‌Examples‌ ‌Are‌ ‌Not‌ ‌Bugs,‌ ‌They‌ ‌Are‌ ‌Features](https://arxiv.org/abs/1905.02175)‌‌ ‌(See‌ ‌Section‌ ‌
> 4)‌ ‌
> [[ICLR’19]‌ ‌Tsipras‌ ‌et‌ ‌al.,‌ ‌Robustness‌ ‌May‌ ‌Be‌ ‌at‌ ‌Odds‌ ‌with‌ ‌Accuracy‌‌](https://arxiv.org/abs/1805.12152) ‌(See‌ ‌Section‌ ‌2.1)‌ ‌
> [[NeurIPS’19]‌ ‌Tramer‌ ‌et‌ ‌al.,‌ ‌Adversarial‌ ‌Training‌ ‌and‌ ‌Robustness‌ ‌for‌ ‌Multiple‌ ‌Perturbations](https://arxiv.org/abs/1904.13000)‌‌ ‌(See‌ ‌Section‌ ‌2.2)‌ ‌
>  ‌
>
> In‌ ‌our‌ ‌paper,‌ ‌we‌ ‌mention‌ ‌that‌ ‌the‌ ‌assumptions‌ ‌made‌ ‌in‌ ‌our‌ ‌work‌ ‌are‌ ‌adapted‌ ‌from‌ ‌Ilyas‌ ‌et‌ ‌al.‌ ‌
> (Section‌ ‌4.1‌ ‌and‌ ‌Appendix‌ ‌A‌ ‌in‌ ‌our‌ ‌paper).‌ ‌In‌ ‌Appendix‌ ‌A‌ ‌of‌ ‌our‌ ‌paper,‌ ‌we‌ ‌have‌ ‌compared‌ ‌our‌ ‌
> assumptions‌ ‌with‌ ‌Ilyas‌ ‌et‌ ‌al.‌ ‌We‌ ‌discussed‌ ‌that:‌ ‌(1)‌ ‌the‌ ‌assumptions‌ ‌made‌ ‌in‌ ‌our‌ ‌work‌ ‌better‌ ‌
> represent‌ ‌the‌ ‌characteristics‌ ‌of‌ ‌real-world‌ ‌image‌ ‌datasets;‌ ‌and‌ ‌(2)‌ ‌our‌ ‌results‌ ‌naturally‌ ‌hold‌ ‌with‌ ‌
> assumptions‌ ‌made‌ ‌in‌ ‌Ilyas‌ ‌et‌ ‌al.,‌ ‌which‌ ‌assumes‌ ‌the‌ ‌presence‌ ‌of‌ ‌a‌ ‌stochastic‌ ‌input‌ ‌feature‌ ‌
> (explained‌ ‌in‌ ‌Appendix‌ ‌A).‌ ‌
>  ‌
>
> On‌ ‌the‌ ‌other‌ ‌hand,‌ ‌we‌ ‌would‌ ‌like‌ ‌to‌ ‌highlight‌ ‌that‌ ‌while‌ ‌the‌ ‌theoretical‌ ‌proofs‌ ‌are‌ ‌important‌ ‌parts‌ ‌
> of‌ ‌the‌ ‌paper‌ ‌to‌ ‌motivate‌ ‌the‌ ‌‌*PROTECTOR* ‌framework,‌ ‌the‌ ‌strong‌ ‌empirical‌ ‌results‌ ‌are‌ ‌also‌ ‌key‌ ‌
> contributions‌ ‌of‌ ‌our‌ ‌work.‌ ‌In‌ ‌Section‌ ‌6,‌ ‌we‌ ‌show‌ ‌that‌ ‌on‌ ‌adversarial‌ ‌examples‌ ‌generated‌ ‌using‌ ‌
> various‌ ‌attack‌ ‌algorithms,‌ ‌the‌ ‌Protector‌ ‌framework‌ ‌outperforms‌ ‌the‌ ‌state-of-the-art‌ ‌defenses‌ ‌by‌ ‌
> large‌ ‌margins.‌ ‌We‌ ‌urge‌ ‌you‌ ‌to‌ ‌kindly‌ ‌also‌ ‌consider‌ ‌the‌ ‌empirical‌ ‌and‌ ‌methodological‌ ‌
> components‌ ‌of‌ ‌our‌ ‌work‌ ‌in‌ ‌your‌ ‌review‌ ‌and‌ ‌subsequent‌ ‌decision.‌    ‌
>  ‌
>
> In‌ ‌our‌ ‌revision,‌ ‌we‌ ‌will‌ ‌use‌ ‌the‌ ‌additional‌ ‌page‌ ‌to‌ ‌move‌ ‌some‌ ‌discussion‌ ‌of‌ ‌the‌ ‌dataset‌ ‌
> assumptions‌ ‌to‌ ‌the‌ ‌main‌ ‌paper.‌ ‌We‌ ‌are‌ ‌happy‌ ‌to‌ ‌provide‌ ‌further‌ ‌clarifications.‌

---

### Official Review · AnonReviewer3 · 2020-10-27
**A good pipeline idea to improve robustness**

**Rating:** 6
**Confidence:** 3

**Review:**

Summary:
This paper proposed a pipeline method which first classify the attack type and then choose the proper predictor for that type. The authors provide both theoretical and experimental proof to support their pipeline method.

Pros:

(1) The idea is good and reasonable. Also, the paper is well written and easy to read. The proof part is clear and natural.
(2) The result of theorem 1 is interesting and it proved the different perturbation types (under this problem setting) is separable with a high probability. Although the setting is a little restrictive, it is still very impressive.

Cons:
(1) The method and analysis only apply to the Lp attack type. It will be good if it can be extended.
(2) Assume that we treat the two pipelines as the whole process (end-end deep network), e.g., the first layers determines the type and then that type will determine which part to activate for training the data points, that is, you have a network with special topology and structure. If the attacker only cares about this input and output of this network (no pipeline here), we should have an attack strategy. How to explain this view using your analysis ? Could we say this is equivalent  to changing the structure of the network  ? From this perspective, could you tell the difference between these two views ?

---

> ### Author Response · Authors · 2020-11-15
> **Multiple perturbation types and the model structure**
>
> Thank you for your insightful feedback on the work. We are glad you liked the clarity of the paper and our theoretical arguments that motivate the empirical results. We address your concerns and questions below, and we will revise our paper to make these points clearer. ‌
>  ‌
>
> ### $\ell_p$ Norm Setting
> We agree that the problem of multiple $\ell_p$ robustness may be a little restrictive, but it is also one of the only settings in which we have a standardized notion of attack and defense benchmarks. We believe that multiple perturbation robustness to other threat models will naturally follow in our *PROTECTOR* framework, as more attack types and individual models robust to them become standardized. However, in the absence of such attacks and defenses, it is difficult to demonstrate the efficacy of the same. We also note that the problem of multiple $\ell_p$ norm robustness has attracted a lot of attention from the community. Apart from the citations in the main paper, there are at least the following submissions at ICLR this year that also study the same problem: ‌
>
> [Towards Defending Multiple Adversarial Perturbations via Gated Batch Normalization](https://openreview.net/forum?id=Utc4Yd1RD_s)
> [Composite Adversarial Training for Multiple Adversarial Perturbations and Beyond](https://openreview.net/forum?id=H92-E4kFwbR)
> [Learning to Generate Noise for Multi-Attack Robustness](https://openreview.net/forum?id=tv8n52XbO4p) ‌
>  ‌
>
> ### Topological Model
> *If the attacker only cares about this input and output of this network (no pipeline here), we should have an attack strategy. How to explain this view using your analysis? Could we say this is equivalent to changing the structure of the network? From this perspective, could you tell the difference between these two views ?* ‌
>  ‌
>
> #### **Attack Strategy**
>
> For both the standard attack (Eq(1)) and the softmax adaptive attack (Eq(4)), our entire pipeline (with two stages) is considered as a single model at test time. The adversary utilizes the knowledge of the input and output pairs from the entire pipeline to frame the attack strategy, as also suggested by you. In fact, the adaptive attack presented in (Eq(4)) further exploits the knowledge of the internal model structure to propagate full gradients through the two stages, and hence produces stronger adversarial examples than standard attacks that only care about model input and output.
>
> We agree that our entire pipeline is essentially equivalent to a topological model, for which the first few layers perform perturbation type categorization, and then specialized layers for different perturbation types follow. To explain why our pipeline is supposed to be more robust to multiple perturbation types, we would like to draw your attention towards the adversarial trade-off discussed in Theorem 2 and represented in Figure 1.b. Specifically, we design this topological structure to establish a trade-off for the adversary between fooling the two stages of the topology, even when it taking a unified view of the model and *not* attacking the two stages separately. ‌
>  ‌
>
> #### **Training the Topological Model**
>
> The key advantage that *PROTECTOR* brings is in the ability to train such a topological model in the first place. Recent work ([[ICML2020] Adversarial Robustness Against the Union of Multiple Perturbation Models](https://arxiv.org/abs/1909.04068)) has demonstrated how naive augmentation of attacks from different perturbation types leads to unstable and inconsistent training when used to train a single robust model, and often suffers from various kinds of gradient masking. On the other hand, as mentioned in Section 5.1, we observe that the training of the perturbation categorizer is stable and consistently achieves good performance.
>
> Further, as also detailed in Section 5.1, these previous models take an extremely large amount of computational resources to train. For instance, the state-of-the-art network for $\ell_\infty$ robustness (with data augmentation) takes nearly 2 GPU days (https://github.com/yaircarmon/semisup-adv). Doing the same with augmentation of data points for different types of $\ell_p$ attacks like in [[NeurIPS’19] Tramer et al., Adversarial Training and Robustness for Multiple Perturbations](https://arxiv.org/abs/1904.13000) will require 3x the time. On the other hand, the *PROTECTOR* framework is able to leverage the strong predictions of pre-existing individually robust models to train a perturbation categorizer in only a few hours.
>
> We are happy to provide any further clarification about the work and thank you for your time.

---

### Official Review · AnonReviewer1 · 2020-10-29
**The paper looks good but some points are unclear to me**

**Rating:** 6
**Confidence:** 4

**Review:**

This paper proposes an ensemble approach to deal with multiple perturbation types. The underlying idea is to train a robust classifier for each perturbation type (i.e., l1, l2, and l-inf) and choose a model to predict based on the decision of a perturbation classifier which is trained to distinguish perturbation types.
The idea is interesting, and the experiments seem solid; however, I am hesitating to give this paper an acceptance decision due to some unclear points as follows:
1.	I have not checked the proof of Theorem 1, but it seems a bit counter-intuitive to me. The reason is as follows. Let’s assume that we are attacking a clean image x. We start from x0 inside both l1 and l-inf balls for example. We further assume that during the process of updating x{t}, we never do any projection onto any ball (i.e., x{t} always lie inside the balls). If so, there is not any difference between x_adv w.r.t l1 and l_inf, meaning that there are possibly a certain number of adversarial examples shared across l1 and l-inf attacks.
2.	In your Theorem 1 (also your experiments), the radius of l1 is eps and that of l-inf is eps/sqrt(d), meaning that the ball of l-inf is a subset of the ball of l1. However, in Section 5.3, you claimed that “The dotted line shows the decision boundary for the perturbation classifier C_adv, which correctly classifies inputs subjected to large ‘1 perturbations δ00 as ‘1 attacks (green), but can misclassify samples with smaller perturbations”. It seems that you reckon l-inf adversarial examples are having more perturbation than l1, do not you?
3.	Theorem 2 is not understandable to me. What is worst-case adversary (this needs to be explained and defined because we can understand this notion in several different ways)? What is delta?
4.	I recommend testing Projector against the attack over a uniform average of Mp in addition to what is doing.
5.	It is encouraging to compare your proposed method against (Francesco Croce and Matthias Hein, 2020 a). You did mention this work, but not compare it in experiments.

---

> ### Author Response · Authors · 2020-11-15
> **Clarifications for Reviewer 1**
>
> Thank you for your detailing your concerns. We are happy to note that you found the idea to be interesting, while being backed up with solid experiments. We attempt to clarify all your concerns in this response, and we will also revise our paper to make these points clearer. ‌
>  ‌
>
> We discuss the **separability of perturbation types** in the common response: [Non-Overlapping Nature of different perturbation regions](https://openreview.net/forum?id=Oe2XI-Aft-k&noteId=V1q5prt8RqE). The response addresses Cons (1,2) in your review.
>
> Following up on this discussion:
> *The radius of l1 is eps and that of l-inf is eps/sqrt(d), meaning that the ball of linf is a subset of the ball of l1*
> While the *magnitude* of the radius of the $\ell_1$ region is larger than the $\ell_\infty$ region, none of the regions subsume the other. It is possible for a point in the $\ell_\infty$ space to exist outside the $\ell_1$ region. Example: Consider a 100-dimensional input $\delta = [0.3,0.3,...,0.3]$. It exists in the $\ell_\infty$ region of $\epsilon_\infty = 0.3$. However, the sum of absolute value of each component of $\delta$, or its $\ell_1$ norm is equal to 30. Hence, the perturbation does not belong to a region of size $\epsilon_1 = 10$. ‌
>  ‌
>
>
> *In Section 5.3, you claimed ...It seems that you reckon l-inf adversarial examples are having more perturbation than l1*
> Our discussion of the perturbation classifier does not utilize this assumption. Instead, we mean that for an $\ell_\infty$ adversary, if the generated adversarial perturbation is large based on the $\ell_\infty$ norm, then the perturbation classifier is likely to correctly classify the adversarial example as $\ell_\infty$. On the other hand, when the generated adversarial perturbation is small based on the $\ell_\infty$ norm, the perturbation classifier may not correctly categorize the perturbation type, because the perturbation may not be recognizable enough. We will revise the description of Section 5.3 to make these points clearer, and we will clarify the meaning of small and large perturbations. ‌
>  ‌
>
>
> ### Theorem 2
>
> #### *Terminologies*
> 1. Worst-case adversary refers to an adaptive adversary that has the full knowledge of the defense strategy, and makes the strongest adversarial decision given the perturbation constraints. We will add the explanation in our revision.
> 2. As described in Equation 3, $\delta$ is the perturbation induced over the original input $x$, such that $x_{adv} = x + \delta$.
> 3. $\alpha, \sigma, d$ are defined in the problem description in Section 4.1. ‌
>  ‌
>
>
> #### *High-Level Summary*
> In Theorem 1, we showed that it is possible to distinguish between the distributions of inputs that were subjected to $\ell_1$ and $\ell_\infty$ perturbations. In Theorem 2, we prove lower-bounds on the robustness of our two-stage pipeline under an adaptive attacker, and show that the adversary faces a trade-off between fooling the perturbation classifier and the second-level models.
>  ‌
>
>
> **What does the theorem claim?**
> Error rate of the *PROTECTOR* model, $P_e$ is less than 0.01 under the considered threat model.
> **What is the problem setting?**
> 1. Data points are sampled from $\mathcal{D}$ defined in Section 4.1.
> 2. Adversarial perturbation regions considered are the $\ell_1$ and $\ell_\infty$ regions.
> 3. Protector pipeline consists of adversarially trained models $M_{1,\epsilon_1}, M_{\infty,\epsilon_\infty}$, and an attack classifier $C_{adv}$.
> 3. The perturbation size $\boldsymbol{\epsilon_1} = \alpha + 2\sigma$ and $\boldsymbol{\epsilon_\infty} = \frac{\alpha + 2\sigma}{\sqrt{d}}$.
>
> For a more detailed description, please refer to the proof sketch of the theorem provided at the beginning of Appendix C. In a subsequent revision, we will utilize the additional page for the main paper to highlight the proof sketch, and provide a more intuitive explanation of the theorem. ‌
>  ‌
>
>
> ### Additional Comparisons
> As noted in the paper, we build on adversarial training baselines (Tramer & Boneh 2019, Maini et. al. 2020). However, Croce & Hein 2020, study provable robustness. As a result, the perturbation sizes considered in their work are significantly smaller than those in empirical works such as ours. For example, in case of the MNIST dataset, while we study empirical robustness at $\epsilon_1$ = 10.0, Croce & Hein study the upper and lower bounds of the robust test error on $\epsilon_1$ = 1.0 (see Table 1 in [Croce & Hein 2020](https://arxiv.org/abs/1905.11213)). They present a range of certified robust error rates, without empirical results as evaluated in our work & the baselines considered. Thus, the two works have different objectives and are not empirically comparable. ‌We will make this point clearer in our revision.
>  ‌
>
> Thank you for your time. We will shortly provide evaluation results on the uniform average of $M_p$ in a succeeding response. In the meantime, please let us know if you have any more questions.

---

> > ### Author Response · Authors · 2020-11-23
> > **Feedback incorporated and more experiments**
> >
> > In our revision, we have incorporated all the changes that you recommended to enhance the clarity of the narrative. A summary of the changes can be found [here](https://openreview.net/forum?id=Oe2XI-Aft-k&noteId=UduYTjNuVJG).
> >
> > ### Attack over a uniform average of $M_p$
> >
> > Thank you for the suggestion. In the table below we present results of the *PROTECTOR* model when naively using a uniform average over the outputs from each individual model. We find that there is nearly a 25% accuracy drop for $\ell_\infty$ and $\ell_2$ PGD attacks, and 4% for sparse $\ell_1$ attack. We observe accuracy drops for both MNIST and CIFAR-10.
> >
> >                           Robust Accuracies (MNIST)
> >
> > | Attack ||| | Protector | Uniform Average | |
> > |-------------------------- |-|-|-|----------------------- |-------------------------- |--- |
> > | PGD $\ell_\infty$ ($\epsilon_\infty$ = 0.3) ||| | 86.7% | 62.8% | |
> > | PGD $\ell_2$ ($\epsilon_2$ = 2.0) |||| 73.5% | 50.4% | |
> > | PGD $\ell_1$ ($\epsilon_1$ = 10.0) |||| 69.3% | 65.3% | |

---

### Official Review · AnonReviewer2 · 2020-10-30
**The good results may be just brought by inadequate attack evaluation.**

**Rating:** 4
**Confidence:** 4

**Review:**

The paper proposes a two-stage defense method to improve the adversarial robustness over different perturbation types. Specifically, it first builds a hierarchical binary classifier to differentiable the perturbation types and then uses the result to guide to its corresponding defense models.  It first proves the different types of perturbations could be separable and the adversary could be weakened to fool the binary classifier. It shows their methods achieve a clear improvement in the experiments.

Pros:
1. The paper is good-written and easy to follow.
2. The proposed idea is interesting.
3. The experiment is detailed and comprehensive.

Cons:
1. There is a major problem in their method. In the last sentence of section 5.2, it says uses the soft relaxation only in generating the adversarial example, but not for inference. It clearly caused the gradient making problem in the adversarial attack later on to test the robustness.  The gradient is blocked before reaching into the binary classifier so that the adversarial attack fails, which I think it is not truly improving the model's robustness.
2. In my opinion, this method is just a dynamic voting based model ensemble. Just take the binary classifier as a voting procedure.  Therefore, the traditional adversarial attacks won't work in general. I would suggest using the soft relaxation in the inference as well for the adversarial attack.
3. Also, the assumption that different norm adversarial examples could be clearly separable might be wrong. You could find the adversarial examples that satisfy both l1, l2 and l_inf constraint by just choosing the \epsilon for every norm differently.

---

> ### Author Response · Authors · 2020-11-12
> **Important ‌Clarification‌ ‌on‌ ‌Evaluation‌**
>
> Thank you for your feedback and insightful comment on the possibility of gradient masking. We would like to clarify an important misunderstanding about the evaluation adequacy in this response, and will provide a detailed discussion on the adversarial example separability in a followup post.
>
> *"In the last sentence of section 5.2, it says uses the soft relaxation only in generating the adversarial example, but not for inference. It clearly caused the gradient making problem in the adversarial attack later on to test the robustness."*
>
> We would like to clarify that in our experiments, the adversary always constructs adversarial examples using the softmax mode in the pipeline, as described in Equation 4. Indeed, the possibility of gradient masking was our primary motivation to include adaptive attacks (using the softmax relaxation), as discussed in Section 5.2. With our adaptive attacks, the gradient is never blocked while generating the adversarial example.
>
> The model applies the “max” mode (described in Equation 1) only to provide the final prediction for generated adversarial examples. In our evaluation, we found a negligible impact of changing to the “softmax” mode for producing the predictions. For example, in case of the APGD($\ell_\infty$, $\ell_2$) attacks on CIFAR-10, our results are as follows:
>
> 								Robust Accuracies
>
> | Attack                    ||| | Max Approach (Eq (1)) | Softmax Approach (Eq(4)) |   |
> |--------------------------   |-|-|-|-----------------------   |--------------------------   |---   |
> | APGD-CE $\ell_2$ ($\epsilon_2$ = 0.5)   ||| | 75.7%                 | 75.6%                    |   |
> | APGD-DLR $\ell_2$ ($\epsilon_2$ = 0.5)    ||| | 76.5%                 | 76.7%                    |   |
> | APGD-CE    $\ell_\infty$ ($\epsilon_\infty$ = 0.03)   ||| | 86.9%                 | 86.9%                    |   |
> | APGD-DLR  $\ell_\infty$ ($\epsilon_\infty$ = 0.03)    |||| 91.8%                 | 91.2%                    |   |
>
>
>
> In the spirit of consistency of the defense irrespective of whether the attack was adaptive or not, we decided to follow Eq (1) for the final accuracy calculation everywhere.
>
> Further, as noted in the paper, we evaluate our models with the most comprehensive set of attacks in the adversarial ML literature, including both gradient-based and gradient-free attacks. We hope that you reconsider your review in light of this clarification. We will address other concerns in a succeeding response, and will revise our submission to make these points clearer.

---

> ### Author Response · Authors · 2020-11-15
> **Separability of perturbation types**
>
> We discuss the separability of perturbation types in the common response here: [Non-Overlapping nature of different perturbation regions](https://openreview.net/forum?id=Oe2XI-Aft-k&noteId=V1q5prt8RqE). The response addresses Con (3) detailed in your review.  Please refer to our [previous response](https://openreview.net/forum?id=Oe2XI-Aft-k&noteId=sIcexq1_Ibq) in this thread for discussion over other concerns.

---

### Author Response · Authors · 2020-11-15
**Non-Overlapping nature of different perturbation regions**

Multiple reviewers have requested for further clarification on the amount of overlap among different perturbation regions. At a high level, while the $\ell_1,\ell_2,\ell_\infty$ perturbation regions do have overlapping regions (such as $\delta=0$), the strongest adversaries for each perturbation type lie on the boundary of the perturbation region, and are largely non-overlapping. In this response, we first provide visualization aid for these regions, then empirically quantify the overlap, and finally provide intuition for Theorem 1 on Separability of perturbation types. We will also revise our paper to add related discussion. ‌
 ‌

### Visualization of $\ell_p$ regions
Consider the setup for MNIST. While there are points in $\ell_\infty$ region with $\epsilon_\infty$=0.3 which are outside of $\ell_2$ region with $\epsilon_2$=2, the opposite is also true (an $\ell_2$ perturbation at $\epsilon_2$=2 can easily change a single pixel by more than 0.3). These radii are in line with prior work, and **one does not subsume the other**. Similar is the case for our theoretical arguments as well. For a visualization of how these perturbation regions are non-overlapping, you may refer to Figure 1 ([[ICLR2020] Provable robustness against all adversarial $l_p$-perturbations for $p\geq 1$](https://arxiv.org/abs/1905.11213)) for a 3-dimensional perspective and Figure 1 ([[ICML2020] Adversarial Robustness Against the Union of Multiple Perturbation Models](https://arxiv.org/abs/1909.04068)) for a 2-dimensional perspective. As we move to high-dimensional spaces, these perturbation regions become increasingly non-overlapping. ‌
 ‌

### Perturbation statistics
To observe how often adversarial perturbations corresponding to one attack type lie within alternate perturbation regions, we empirically quantify the overlapping regions in case of PGD attacks on the MNIST dataset. In the following, we report the **percentage overlap**, and the **range of the norm** of perturbations in the alternate perturbation region for any given attack type -- when attacking a vanilla model. The observed overlap is 0% in all cases. A similar conclusion holds for CIFAR-10 as well. We will add the full analysis in our revision.

								Adversarial Overlap on MNIST (Empirical)


| Attack                    ||| | $\ell_\infty < 0.3$ |  $\ell_2 < 2.0$ |  $\ell_1 < 10$ |   |
|--------------------------   |-|-|-|-----------------------    |------------------   |--------   |---   |
| PGD $\ell_\infty$ ($\epsilon_\infty$ = 0.3)    |||| 100%   |     0% (3.67 - 6.05)     | 0%  (54.8 - 140.9)    |   |
| PGD    $\ell_2$ ($\epsilon_\infty$ = 2.0)   ||| | 0% (0.40 - 0.86)       | 100%  | 0% (11.2 - 24.1)    |   |
| Sparse-PGD $\ell_1$ ($\epsilon_1$ = 10)    |||| 0%    (0.70 - 1.0)       | 0% (2.08 - 2.92)      |   100%     |   | ‌
 ‌

### Intuition behind Theorem 1
*R1: Let’s assume that we are attacking a clean image x ..... there are possibly a certain number of adversarial examples shared across l1 and l-inf attacks.*
Indeed, there exist regions that are common to both $\ell_1$, $\ell_\infty$ norm bounded spaces. However,
1. Adversaries in such regions can be correctly classified by both the second level models ($M_1$, $M_\infty$). Therefore, these attacks are futile for our paradigm, since the *PROTECTOR* pipeline is robust to them. (In fact, it is an advantage of the pipeline.)
2. The strongest adversaries generally stay at the perturbation boundary. This is because the adversarial objective is typically to maximize the loss of prediction of the victim model (during optimization steps). These regions are generally non-overlapping, and we present a detailed discussion under the previous sub-heading.

Based on the above observations, we show in Theorem 1 that the worst-case adversaries against a vanilla model $M$ are easily separable, with an assumption of perturbation sizes explained below.‌
 ‌



As elaborated in Appendix C.2, we assume that the perturbation size $\epsilon_1$ is sufficient to stage effective attacks against $M_\infty$ model; meanwhile, the perturbation size $\epsilon_\infty$ enables the  $\ell_\infty$ adversary to perform similarly effective attacks against $M_1$ model. The rationale is that if the $\epsilon_p$ adversary already could not attack the alternative model $M_q$ (p \neq q), while  $\epsilon_q$ adversary is able to attack $M_p$, then it means that $M_q$ is already robust to the union of $\epsilon_p$ and $\epsilon_q$ attacks. In our work (and also in the literature of adversarial robustness against multiple attack types) we focus on the scenario where achieving the robustness against a single perturbation type does not trivially imply the robustness against other perturbation types. In fact, prior works have noted that it often hurts the robustness to other attacks. Therefore, we make the above-mentioned assumption to reflect such scenarios. We will make these points clearer in our revision.

---

### Author Response · Authors · 2020-11-23
**Summary of Revision**

Based on the reviews, we have revised our submission to clarify reviewers’ confusions and added additional experiments to provide further evidence.

1. **On Perturbation Statistics**: Multiple reviewers have asked about the overlap among different perturbation types. We added a new section (**Section 6.2**) to discuss the empirical overlap of adversarial perturbations between different norms, and we observe that the overlap is 0% when attacking the vanilla model. We further added more discussion in **Appendix G**, where we show how this overlap varies when attacking *PROTECTOR*.
2. **Theorem 1**: in Section 4.2, we have included a proof sketch to further explain the theorem.
3. **Theorem 2**: in Section 4.3, we have added some discussion on overlapping perturbation regions, and we have also added a proof sketch to further explain the theorem.
4. **Additional Comparison** with Croce and Hein 2020: We have made the distinction between our (empirical) work and their (certified) work clearer in the Related Works.
5. **Theoretical Assumptions** about Dataset Distribution: In Section 3, we have added the discussion about the differences between our problem setting and that chosen by Ilyas et. al. 2018, and further motivated why our assumptions are more realistic and applicable to real-world datasets.
6. **Section 5.2**: We have made it clear that the gradient steps are taken using the adaptive approach (hence not leading to gradient masking), and only the final forward-propagation is done using Equation 1. In Appendix H.4, we have also added comparative results on the effect of changing the method of aggregating predictions from different $M_p$ models.
7. **Section 5.3**: The meaning of small and large perturbations has been clarified.

We once again thank all the reviewers for their feedback that helped enhance the clarity of the draft.

---

### Decision · Program_Chairs · 2021-01-07
**Final Decision**

**Decision:**

Reject

**Comment:**

The paper proposes a model to defend against multiple lp norm attacks by classifying those attacks. The reviewers raised several concerns about the methodologies. Furthermore, it's not clear how the proposed algorithm can deal with an unseen attack (e.g., only trained on l1, l_infty attacks but encounter l2 attack in the testing phase). The assumption that the attack types are known beforehand is restricted.